

# Coupling saturated and unsaturated flow: comparing the iterative and the non-iterative approach

Natascha Brandhorst[1], Daniel Erdal[2,3], and Insa Neuweiler[1]

[1]Institute of Fluid Mechanics and Environmental Physics in Civil Engineering, University of Hannover, Germany
[2]Center for Applied Geoscience, University of Tübingen, Germany
[3]Tyréns AB, Lilla Badhusgatan 2, 411 21 Göteborg, Sweden

**Correspondence:** Natascha Brandhorst (brandhorst@hydromech.uni-hannover.de)

**Abstract.** Fully integrated three dimensional (3D) physically based hydrologic models usually require many computational resources. For many applications, simplified models can be a cost effective alternative. 3D models of subsurface flow are often simplified by coupling a 2D groundwater model with multiple 1D models for the unsaturated zone. The crucial part of such models is the coupling between the two model compartments. In this work we compare two approaches for the coupling. One is

iterative and the 1D unsaturated zone models go down to the impervious bottom of the aquifer and the other one is non-iterative and uses a moving lower boundary for the unsaturated zone. In this context we also propose a new way of treating the specific yield, which plays a crucial role in linking the unsaturated and the groundwater model. Both models are applied to three test cases with increasing complexity and analyzed in terms of accuracy and speed compared to fully integrated model runs. The non-iterative approach is faster while the iterative approach is more accurate and robust. Besides, for the iterative coupling

method a calibration of the specific yield is not needed.

## 1   Introduction

Three dimensional (3D) hydrologic models for subsurface flow are an important tool to gain a better understanding of the processes in a hydrologic system and to estimate the impact of possible future changes to it. However, these models tend to be very computationally demanding especially when applied to settings on large length scales that require a fine grid resolution

(Kollet et al., 2010). Therefore, a lot of approaches to reduce the complexity of such models have been developed. One very common approach is to divide the model domain into two components: the saturated (groundwater) part and the unsaturated part. This is often complemented by the assumption that flow in the unsaturated part occurs predominantly in vertical direction. Thus, the unsaturated zone, which is usually the more complex one due to the strong nonlinearity of Richard's equation, can be represented by one or multiple one dimensional soil columns that can be solved independently from each other. Groundwater

flow is then either modeled as 3D or as 2D depth integrated flow. The coupled model can then be seen as 2.5D or quasi 3D.

In this work we focus on phreatic aquifers where the crucial part is the coupling of the two model components. Usually, they are linked by the exchange of information on groundwater recharge and the position of the groundwater table. This can be done either in an iterative manner to account for the feedback between the two models (Shen and Phanikumar, 2010; Zhu et al., 2011; Kuznetsov et al., 2012; Xie et al., 2012; Xu et al., 2012; Mao et al., 2019; Zeng et al., 2019) or in a non-iterative





manner without feedback (Pikul et al., 1974; Yakirevich et al., 1998; Beegum et al., 2018; Erdal et al., 2019). Both methods have been applied successfully. While iterative coupling in general yields a higher accuracy, non-iterative coupling requires less computational ressources (Zeng et al., 2019).

Differences among the coupling strategies exist also in terms of the spatial extent of the unsaturated zone model. Mostly, the unsaturated zone is modeled to span from the groundwater table to the land surface (Pikul et al., 1974; Yakirevich et al.,
1998; Zhu et al., 2011; Kuznetsov et al., 2012; Xie et al., 2012; Zhu et al., 2012; Renard and Tognelli, 2016; Erdal et al., 2019; Zeng et al., 2019). As the groundwater table is moving with time, a remesh is needed after each coupling time step. Others assign a small overlap of the two model compartments which allows for a better calculation of the recharge flux (Xu et al., 2012; Beegum et al., 2018; Mao et al., 2019). A third approach is to consider 1D soil columns that reach down to the bottom of the groundwater domain (Shen and Phanikumar, 2010). Thus, the two compartments overlap in the entire saturated zone. The
drawback of this approach is that the size of the unsatured model grid is increased. The advantage is that lateral fluxes in the groundwater can be passed directly to the unsaturated zone model and no remeshing is needed.

All these different coupling strategies have been tested successfully against real data or fully integrated 3D models. One has to keep in mind that there are limitations to the applicability that are not related to the coupling strategy but the general 2.5D or quasi 3D approach. Due to the 1D representation of the unsaturated zone, these models cannot be applied in the presence of
large lateral fluxes in the unsaturated zone which might be the case in steep hillslopes. Furthermore, they neglect lateral water movement in the capillary fringe which can play a significant role.

One shared problem of these coupled models is due to the fact that the exchange quantities, e.g. recharge and water table position, are usually assumed to be constant over one coupling time step. This might lead to non-consistent model states when using a coupling scheme without feedback. While the consistency of the two model compartments is not necessary when the
compartments are spatially separated, it is fundamental when they overlap. The easiest way to prevent these inconsistencies is to choose a small enough coupling time step, but this might lead to unfeasible compute times. Beegum et al. (2018) overcame this problem by updating the pressure head profile in the unsaturated zone after each time step in such a way that the new groundwater table position and the current recharge flux are both matched. By this procedure they prevent sudden in- or outflow of the unsaturated zone due to a stepwise changing groundwater table position. However, this requires the application of an
optimization routine which can be rather time consuming. With an iterative coupling scheme, consistency can be achieved, but there still remains an error due to neglection of the temporal changes in the exchange information. Zeng et al. (2019) use linear extrapolation of the groundwater table position to reduce this error. Zhu et al. (2012) solve the two models jointly linked via the pressure head values at the interface. However, this is only possible if the groundwater model is 3D and the time steps in both compartments are equal. Often, the time step of the groundwater model is chosen to be larger than in the unsaturated model
where smaller time steps are needed to achieve convergence for the nonlinear system of equations (Beegum et al., 2018).

Regarding the model consistency, another important factor is the quantification of the specific yield. It relates the fluctuations of the groundwater table to the amount of water being added or released, thus accounting for the neglected unsaturated zone. To get a consistent solution in both model parts in terms of water table position and mass conservation, it is obvious that a correct estimation of the specific yield in the groundwater model is needed.





However, the specific yield is not easy to be quantified as it depends on soil properties, change of the groundwater table and the water content profile above the groundwater table (Pikul et al., 1974; Sophocleous, 1985; Crosbie et al., 2005; Hilberts et al., 2005). The common approach to assign a temporally and often even spatially constant value is clearly a strong simplification in most applications. Beegum et al. (2018) performed a small sensitivity analysis on this parameter and detected a strong influence on the dynamics of the groundwater table position. Some research was dedicated to getting a better estimation of

this value in the context of groundwater modeling (Hilberts et al., 2005) and coupled saturated-unsaturated zone modeling (Pikul et al., 1974; Xu et al., 2012; Zeng et al., 2019). Hilberts et al. (2005) presented an analytical expression for the specific yield depending on the groundwater table position and the soil hydraulic parameters assuming hydrostatic equilibrium in the unsaturated zone. A similar approach was used by Crosbie et al. (2005) who propose a soil and depth dependent formulation of the specfic yield. Pikul et al. (1974) were the first to use a dynamic heterogeneous specific yield for linking the 1D unsaturated

zone model with a 2D Boussinesq equation for the groundwater. They calculate it as the difference between saturated water content and the minimum water content at a specified depth. This approach was criticized by Vachaud and Vauclin (1975), who could confirm the dependency of the specific yield on space and time with physical experiments, but did not agree with its ambiguous definition proposed in the work by Pikul et al. (1974). Xu et al. (2012) implemented three different methods to define the specific yield when coupling the SWAP model (Kroes and Van Dam, 2003) with MODFLOW-2000 (Harbaugh

et al., 2000) but do not give a comparison of these methods or a suggestion. Zeng et al. (2019) distinguish between a small-scale and a large-scale specific yield. The small-scale specific yield is calculated from the unsaturated zone model as the change of water content up to a user specified elevation above the groundwater table, the large-scale specific yield is the commonly used parameter of the groundwater model. The difference between those two is used to correct the recharge flux. So the quantification of the specific yield remains an issue which is not yet fairly solved.

In this work we compare two different coupling methods for linking 1D unsaturated flow with 2D depth integrated groundwater flow. Both are quite simple, straight forward to implement and do not build on any additional software. One of them can keep the consistency of the two model compartments while still being fast and efficient whereas the other one is essentially designed to be very fast. The latter approach builds on the setup developed by Erdal et al. (2019) and uses a non-iterative, fast, but also rather approximate, coupling scheme, in which the unsaturated columns vary individually in length over the sim-

ulation time. Hence, each column is resized every time step to fit the distance between groundwater table and surface. In the other method the unsaturated zone model covers the entire depth to the bottom of the groundwater domain and is coupled in an iterative manner to the groundwater model. Thus, the solution in both model compartments is guaranteed to be consistent. A new way of calculating the specific yield during the iteration is introduced and passed as coupling information between the model components. This prevents a prior calibration of this parameter and can account for its temporal and spatial variability.

We perform a sensitivity analysis to examine whether the calculated specific yield behaves reasonably and to identify those parameters which have a high impact on the groundwater dynamics in the coupled model. The number of parameters that need to be calibrated can thus be reduced saving computation time. As the equations describing unsaturated flow are nonlinear, a global sensitivity analysis, where the sensivity is measured over a larger (predefined) range of parameter values, is preferable. As global sensitivity metric we choose activity scores based on active subspaces introduced by Constantine and Diaz (2017).





This is a rather new method that has been given increasing attention since its publication (Palar and Shimoyama, 2017; Fritz et al., 2019; Leon et al., 2019) and has also been used with subsurface models (Erdal and Cirpka, 2019). The idea of the method is to find the most influential directions in parameter space, in which the directions are linear combinations of the individual parameters, so called active variables.

The remainder of this paper is structured as follows: The coupled model with the two different coupling strategies as well as the activity scores used for the sensitivity analysis are described in detail in the following section. Afterwards the used test cases are introduced. We then present and discuss the results of the coupling scheme comparison and the sensitivity analysis. In the final part we give a summary of our experiments and conclusions.

## 2 Methods

### 2.1 Governing equations

The coupled model consists of two components, namely groundwater and unsaturated zone. Groundwater flow is modeled by the 2D Boussinesq equation for an unconfined aquifer

$$S_y \frac{\partial h}{\partial t} = \nabla \cdot (\overline{K_s}(h - z_0)\nabla h) + R \tag{1}$$

where $S_y$ [−] is specific yield, $h$ [m] is hydraulic head, $t$ [s] is time, $\overline{K_s}$ [ms$^{-1}$] is the depth averaged saturated hydraulic conductivity, $z_0$ [m] is the bottom elevation of the aquifer and $R$ [ms$^{-1}$] is groundwater recharge.

The 1D Richards equation is used to describe flow in the unsaturated zone columns:

$$S(h_p)S_s\frac{\partial h_p}{\partial t} + \frac{\partial(S(h_p)\phi)}{\partial t} = \nabla \cdot (K(h_p)\nabla(h_p + z)) + Q \tag{2}$$

where $S$ [−] is saturation, $S_s$ [m$^{-1}$] is specific storage, $h_p$ [m] is pressure head, $\phi$ [−] is porosity, $K$ [ms$^{-1}$] is the unsaturated hydraulic conductivity, $z$ [m] is the geodetic height above the bottom of the aquifer and $Q$ [s$^{-1}$] is a source/sink term. The relation between pressure head, saturation and unsaturated hydraulic conductivity is represented by the van Genuchten-Mualem model (Van Genuchten, 1980):

$$S(h_p) = \begin{cases} S_r + \dfrac{S_{sat} - S_r}{[1 + (\alpha|h_p|)^n]^m} & \text{if } h_p < 0 \\ S_{sat} & \text{if } h_p \geq 0 \end{cases} \tag{3}$$

$$K(h_p) = \begin{cases} K_s S_e^{0.5}\left[1 - \left(1 - S_e^{1/m}\right)^m\right]^2 & \text{if } h_p < 0 \\ K_s & \text{if } h_p \geq 0 \end{cases} \tag{4}$$





where

$$S_e = \frac{S - S_r}{S_{sat} - S_r} \tag{5}$$

$[-]$ is the effective saturation and the model parameter $m$ $[-]$ is usually given by $m = 1 - 1/n$. The model parameters $\alpha$
$[\mathrm{m}^{-1}]$ and $n$ $[-]$ are related to the pore-size distribution, $S_{sat}$ and $S_r$ $[-]$ are the saturated and residual saturation, respectively.

## 2.2 Coupling

### 2.2.1 General approach

The coupled model consists of a 2D depth averaged model for horizontal groundwater flow (Eq. 1) and multiple 1D models for
vertical unsaturated flow (Eq. 2). Each 1D model is located at a cell of the groundwater model grid. To reduce the computational
burden not every groundwater grid cell has a 1D model. Instead, several grid cells of the groundwater model can be lumped
into zones that are assigned one 1D soil column. A schematic example of the model grid is shown in Fig. 1. The recharge that
is calculated from the unsaturated zone model and passed to the groundwater model is assumed to be constant within each
zone. Therefore, the definition of the zones should be geared to land use, depth to groundwater table or other factors that affect
the recharge flux as also done by Xu et al. (2012), Renard and Tognelli (2016) and Erdal et al. (2019) and investigated by Zhu
et al. (2012) and Zeng et al. (2019).

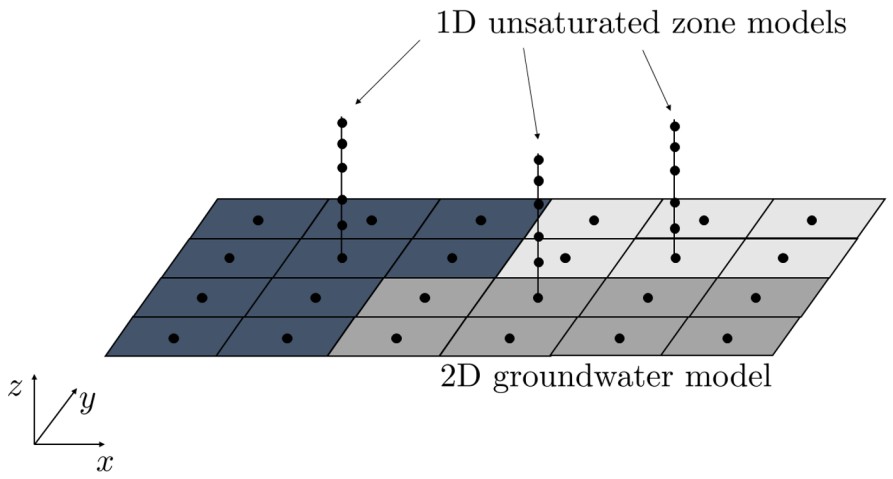

**Figure 1.** Example of the coupled model grid with 3 recharge zones (white, gray and black).

Both components are discretized using finite volumes and an implicit Euler time integration. The nonlinear systems of
equations are solved using the Newton Raphson scheme and line search. For the groundwater model a rectangular grid with
uniform grid spacing is used. The 1D grid for the unsaturated zone can be non-uniform.





The boundary conditions on the lateral sides of the groundwater model can be either Neumann or Dirichlet boundary conditions. The upper boundary condition of the unsaturated zone models is a Neumann boundary representing net flux from precipitation and evapotranspiration. The feedback from the groundwater model to the unsaturated zone models as well as the lower boundary condition of the unsaturated zone models depend on the coupling method described in the following sections.

     The time step size of the unsaturated zone model is smaller than that of the groundwater model to guarantee convergence
in the former but also numerical efficiency for the full model. The coupling time step, which defines the interval when the two model components exchange their information, is equal to the time step size of the groundwater model. All exchanged information such as groundwater recharge etc. is assumed to be constant during one coupling time step.

     Both components as well as the coupling routine are implemented in Matlab.

### 2.2.2   Non-iterative method

The first coupling method aims at providing a fast and iterative free, but also rather approximate, solution to the numerical problem. A previous version of the model is published in Erdal et al. (2019). In this method, specific yield is a parameter of the groundwater model that the user needs to specify during model setup. Each unsaturated zone column is set up to range from the groundwater table to the land surface, with a Dirichlet boundary $h = 0\,\mathrm{m}$ at the bottom. A coupling time step is then performed as:

1. Run all unsaturated 1D column models for the coupling time step, collect the computed recharge (i.e. flux leaving over the bottom boundary) and interpolate the 2D map of groundwater recharge.

     2. Run the 2D groundwater model for the coupling time step with the computed recharge.

     3. For each 1D column, resize all cells by a uniformed ratio computed such that a resized grid fits the new groundwater table to land surface distance at the column location. All other properties of the model are kept as they are, including
155         saturation and hydraulic head.

     4. Compute the amount of water lost (or gained) in each unsaturated model by resizing the grid.

     5. Add (or subtract) a ratio $r$ of this water to the recharge computed in the next time step.

     6. Move to the next coupling time step and repeat from point i.

The ratio $r$ can in principle range from 0 to 1. At 1 the coupled model is locally mass conservative, as no water is lost during
the grid-resizing. However, a local mass-conservation in a non-iterative coupling method may not be wanted. Reasons for this are for example (1) the groundwater table rise or fall is also effected by lateral flows, (2) the added or subtracted water is only added in the next time step, (3) the dynamics around the rising or falling groundwater table is only captured in rough terms, and, (4) the column resizing is distributed across the full column. In this work we compute $r$ as the ratio between the incoming water from the 1D column ($R\Delta t_c$ [m]), the 1D column porosity ($\phi$ [$-$]) and the change of groundwater table elevation ($\Delta H_{GW}$





[m]):

$$r_i = \min\left\{\left|\frac{R_i\Delta t_c/\phi_i}{\Delta H_{GW,i}}\right|, 1\right\} \tag{6}$$

where $i$ is the index of the column, $\Delta t_c$ [s] the coupling time step and $r$ is limited to a maximum value of 1. The current formulation gives an indication to what extent the change of groundwater elevation can be attributed to the incoming groundwater recharge. If recharge is the main contribution, the water lost in the resizing could be assumed to have entered the groundwater, while if the recharge is small but the change in groundwater table large, we find it more likely that the water may still be in the unsaturated zone (e.g. the unsaturated zone is really compacted by a rise of groundwater levels). We note that this is an ad-hoc solution, and that it differs to the one presented in Erdal et al. (2019), where $r$ was simply set to 0. However, numerical tests show good stability and better detailed performance when $r$ is computed as in this work.

### 2.2.3 Iterative method

The second method couples the groundwater and the unsaturated model parts iteratively. Here, the 1D soil columns span the entire soil thickness from the impervious bottom of the aquifer up to the land surface. Thus, the 1D and the 2D model overlap in the saturated part. The bottom boundary condition of the 1D columns is then a no-flow boundary.

The groundwater recharge $R$ [ms$^{-1}$], that is passed from the unsaturated zone models to the groundwater model, is defined - similar as in Crosbie et al. (2005) - as the amount of water being added to or substracted from the groundwater over time and calculated from each unsaturated flow model as

$$R^\nu = \frac{\Delta H^1_{UZ} \cdot S^\nu_y}{\Delta t_c} \tag{7}$$

where $H_{UZ}$ [m] is the position of the groundwater table in the unsaturated zone model, $S_y$ [−] is the specific yield, $\Delta t_c$ [s] is the coupling time step and the superscript ($\nu$) the iteration counter.

The information passed from the groundwater model to the unsaturated zone models are the lateral fluxes $Q_{lat}$ [ms$^{-1}$] into or out of the columns cells. They are calculated at the locations where the unsaturated and groundwater models overlap as the amount of water being added to or substracted from the groundwater over time minus recharge:

$$Q^\nu_{lat} = \frac{\Delta H^{\nu-1}_{GW} \cdot S^{\nu-1}_y}{\Delta t_c} - R^{\nu-1} \tag{8}$$

with $H_{GW}$ [m] being the position of the groundwater table in the groundwater model. This cumulative flux is spread over the saturated part of the corresponding 1D model taking into account the variable cell sizes of the 1D grid:

$$q_{lat,i} = Q_{lat} \cdot \frac{\Delta z_i}{\sum_j \Delta z_j} \quad \text{with} \quad z_i, z_j \leq H_{UZ} \tag{9}$$





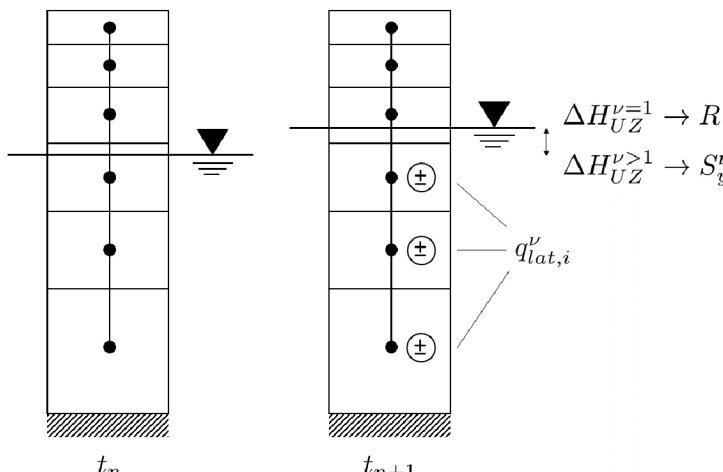

**Figure 2.** Illustrative example of the time step integration for one 1D model in the iteratively coupled model.

The local flux $q_{lat,i}$ [ms$^{-1}$] is given as a source/sink term to the $i$-th cell of the 1D model. This procedure is shown exemplarily for one 1D model in Fig. 2.

In this approach, the position of the groundwater table in both models is not used as exchange information but as closure criterion for the iteration. With the lateral fluxes from groundwater flow being accounted for in the unsaturated zone models, the calculated groundwater table position at the next time step should be equal in both models. However, this is only the case if the specific yield is chosen to correctly represent the behaviour of the unsaturated zone in the groundwater model. As was already discussed in the introduction, the specific yield is dynamic. Therefore it needs to be determined at every time step. In the literature, different approaches of how to define and calculate the specific yield can be found. Most of them lack the dependency on the fluctuations of the groundwater table $\Delta H$. The method proposed by Zeng et al. (2019) takes this into account but requires the definition of a threshold elevation above the groundwater table. Here, we propose a method that uses the specific yield as an adjustment parameter and calculate it depending on $\Delta H$ without need of further user-defined parameters:

$$S_y^\nu = \frac{Q_{lat}^\nu \cdot \Delta t_c}{\Delta H_{UZ}^\nu - \Delta H_{UZ}^1} \tag{10}$$

It represents the ratio of the lateral fluxes and the corresponding change of the groundwater table position in the unsaturated zone model. In our case this is equivalent to relating all incoming fluxes ($Q_{lat}^\nu + R^\nu$) to the total change of the groundwater table position ($\Delta H_{UZ}^\nu$). Using this value in the groundwater model the same lateral fluxes will cause the same response of the groundwater table in the unsaturated and in the groundwater model. As the altered specific yield may lead to different lateral fluxes and consequently to a different specific yield, several iteration steps are needed to achieve a consistent solution in both





model compartments. Note that if $\Delta H^{\nu}_{UZ} - \Delta H^1_{UZ} = 0\,\mathrm{m}$, then there are no lateral fluxes which means that there is no iteration
required and the specific yield stays at its current value.

The integration of the coupled model to the next time step can be thus summarized as follows:

1. Run all unsaturated 1D column models for the coupling time step with $q_{lat,i} = 0\,\mathrm{ms}^{-1}\ \forall i$, calculate the computed recharge according to Eq. 7 and interpolate the 2D map of groundwater recharge $R^1$.

2. Run the 2D groundwater model for the coupling time step with the computed recharge $R^1$.

3. Calculate the lateral flux source/sink terms $q^{\nu}_{lat,i}$ for the 1D models (Eq. 9).

4. Run all unsaturated 1D column models for the coupling time step with $q^{\nu}_{lat,i}$, calculate the computed recharge according to Eq. 7 and interpolate the 2D map of groundwater recharge $R^{\nu}$.

5. Calculate and interpolate the 2D map of specific yield $S^{\nu}_y$ using Eq. 10.

6. Run the 2D groundwater model for the coupling time step with the updated recharge $R^{\nu}$ and specific yield $S^{\nu}_y$.

7. Repeat steps (iii) to (vi) until $|H^{\nu}_{UZ} - H^{\nu}_{GW}| \leq \varepsilon_H$.

8. Move to the next coupling time step and repeat from point i.

Figure 3 shows the iterative coupling procedure. Note that an assumption is made in this approach. The source/sink terms $q_{lat,i}$ have an effect on the recharge ($R(Q_{lat} = 0) \neq R(Q_{lat} \neq 0)$), which due to the nonlinearity of Richards equation cannot be quantified. This effect is assumed to be negligible and the fluctuations due to recharge ($\Delta H^1_{UZ}$) are kept constant during the
iterations whereas the remainder of the total fluctuations ($\Delta H^{\nu}_{UZ} - \Delta H^1_{UZ}$) are assigned entirely to the lateral fluxes. Besides, as groundwater recharge and specific yield are calculated from the unsaturated zone models, they are constant for all cells of the groundwater model that belong to the zone assigned to that unsaturated zone model. This leads to a simplified recharge and specific yield pattern but is necessary to keep the model computationally efficient.

## 2.3 Activity scores

We perform a global senstivity analysis to identify the most influential parameters for the groundwater dynamics in the two coupled models. As metric we use activity scores. Therefore we define the vector of $m$ model parameters $\boldsymbol{x}$ normalized to a range of $[0,1]$ with the probability density function $\rho(\boldsymbol{x})$ of the parameters and the scalar model output $f(\boldsymbol{x})$. The parameters and the model output $f$ are defined in Sections 3.4 and 4.4, respectively. The active subspace of $f$ is defined by the eigenvectors of the matrix

$$\mathbf{C} = \int \nabla f(\boldsymbol{x}) \nabla f(\boldsymbol{x})^T \rho(\boldsymbol{x}) d\boldsymbol{x} = \mathbf{W}\Lambda\mathbf{W}^T, \tag{11}$$





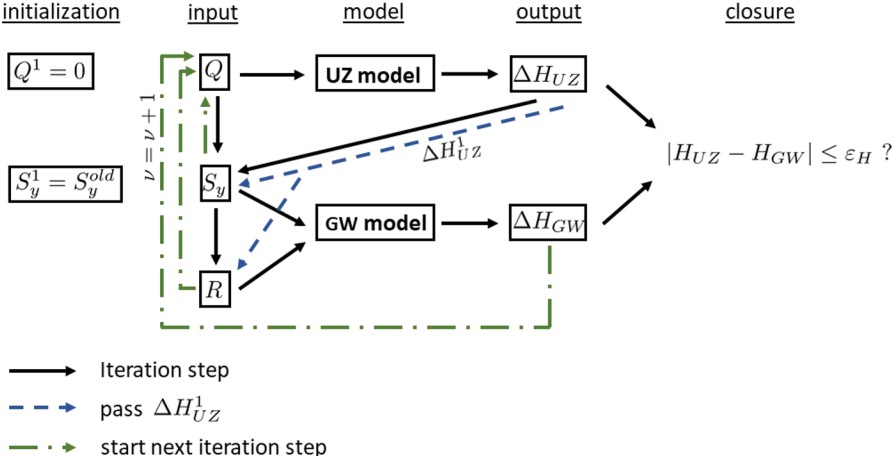

**Figure 3.** Flow diagram for the iterative coupling.

where $\mathbf{W}$ is the matrix of eigenvectors and $\Lambda$ is the diagonal matrix of the corresponding eigenvalues $\lambda_i$ in decreasing order. If there exists a $\lambda_n$ that fulfills $\lambda_n \gg \lambda_{n+1}$, the individual sensitivity for the $i$-th parameter can be calculated as

$$a_i = a_i(n) = \sum_{j=1}^{n} \lambda_j w_{ij}^2 \tag{12}$$

where $\lambda_j$ is the $j$-th eigenvalue and $w_{ij}$ the value corresponding to the $i$-th parameter in the $j$-th eigenvector. This metric is

called the activity score. For a transient problem, the activity score of a parameter can change with time, leading to a transient activity score $a_i(t)$.

In this work, the activity score is computed independently for each time step to create a transient score and normalized such that the sum of all scores is equal to one:

$$NAS_i(t) = \frac{a_i(t)}{\sum_{i=1}^{m} a_i(t)}. \tag{13}$$

As proposed in Constantine et al. (2015), a Monte Carlo method is used to compute the matrix $\mathbf{C}$:

$$\mathbf{C} \approx \frac{1}{M} \sum_{i=1}^{M} \nabla f(\boldsymbol{x}_i) \nabla f(\boldsymbol{x}_i)^T \tag{14}$$





where $M$ samples are drawn independently from $\rho(\boldsymbol{x})$. Following the suggestion by Erdal and Cirpka (2019), a second order polynomial depending on the input parameters is fitted to the data $f(\boldsymbol{x})$ using standard multiple regression, from which the gradient $\nabla f$ can be computed easily:

$$f(\boldsymbol{x}) \approx b_0 + \sum_{i=1}^{m} b_i x_i + \sum_{i=1}^{m}\sum_{j=1}^{m} b_{ij} x_i x_j. \tag{15}$$

## 3 Test cases

We test and compare the above described coupling methods using three different test cases. The first two test cases are taken from Beegum et al. (2018) where the second one was originally set up by Morway et al. (2013). As these are rather artificial, we added a third test case which is more complex and more realistic. The first test case is a flow model in a bucket setup where all boundary sides are closed. It represents a 1D flow problem consisting of two homogeneous soil layers with dynamic precipitation evapotranspiration (PET) data. It is used to investigate the coupling in the absence of any lateral fluxes where fluctuations of the groundwater table are only due to groundwater recharge. The second test case is conceptualized with a 3D model domain. However, as all parameters are constant over the length of the domain, it is effectively a 2D flow problem. Here again the soil is homogeneous, but it contains lateral fluxes. The third test case is similar to the second but the dimensions and the precipitation are modified such that the groundwater table falls during a certain period and the soil is made heterogeneous. This test case is a true 3D flow problem. The main features of the test cases are summarized in Table 1. For the 1D bucket model, one 1D soil column that is not coupled to groundwater is solved for comparison. The results of the other two test cases are compared to those obtained from a fully integrated 3D model set up with Parflow (Kollet and Maxwell, 2006).

**Table 1.** Main features of the three test cases.

|  | 1D flow | 2D flow | 3D flow |
|---|---|---|---|
| soil heterogeneity | no | no | yes |
| lateral fluxes | no | yes | yes |
| rising and falling groundwater table | no | no | yes |
| simplified $R$ and $S_y$ pattern | no | no | yes |

For the global sensitivity analysis a fourth test case is created which is mainly a combination of the second and the third test case. They are combined and modified in such a way that the groundwater table rises and falls, the simulation is long enough to finish the model spin up and parameters are homogeneous within each model compartment.





### 3.1 Test case 1: 1D flow

This model is taken from Beegum et al. (2018) and covers a domain of $1\,\mathrm{m} \times 1\,\mathrm{m} \times 10\,\mathrm{m}$. All lateral boundaries are no-flow boundaries. The net flux of daily varying precipitation and potential evapotranspiration shown in Fig. 4 is applied as a Neumann boundary at the soil surface. This data was collected at a weather station in Gdansk, Poland in 2011. The initial condition assumes a hydrostatic pressure profile corresponding to an initial position of the groundwater table at $3.95\,\mathrm{m}$ below the surface. In the upper $3.5\,\mathrm{m}$ of the soil the pressure head was set to $-0.283\,\mathrm{m}$. The soil consists of two layers, a sandy soil in the top $2.5\,\mathrm{m}$ and a loamy sand soil underneath. The soil hydraulic parameters for the two layers are given in Table 2. Since the groundwater table stays below the interface to the sandy soil layer during the entire simulation, the groundwater model is assigned the loamy sand parameters.

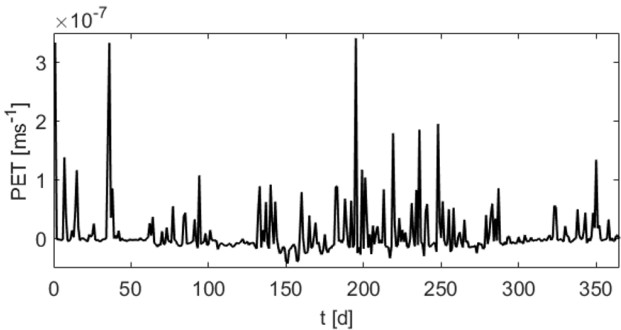

**Figure 4.** Net precitpitation and evapotranspiration used for the 1D flow model. Positive values indicate inflow, negative values outflow.

The total simulation time is one year. The 1D model used for comparison uses an adaptive time stepping scheme with a minimum value of $1\,\mathrm{s}$ and a maximum value of $1\,\mathrm{h}$. The grid is uniform with $1000$ cells of $0.01\,\mathrm{m}$ height. The same spatial resolution is used for the 1D models in the iteratively coupled model. In the non-iterative model a grid consisting of $257$ cells with variable grid size is used. As the 1D grids are remeshed after every time step in the non-iterative model, it is not necessary to use the same spatial discretization as in the iterative model. The groundwater domain is divided into a $2 \times 2$ grid. Each groundwater cell is assigned a 1D model. The coupling time step as well as the time step of the groundwater model is set to $1\,\mathrm{d}$ for both coupling methods. The time step for the 1D models is adaptive with a minimum value of $1\,\mathrm{s}$ and a maximum value of $1\,\mathrm{d}$.

### 3.2 Test case 2: 2D flow

The dimensions of this model (Morway et al., 2013; Beegum et al., 2018) are $8000\,\mathrm{m} \times 4000\,\mathrm{m} \times 15\,\mathrm{m}$. Since there is no variability along the $8000\,\mathrm{m}$ side, flow is effectively 2D in this test case. There is a constant slope of $0.1\,\%$ at the $4000\,\mathrm{m}$ side. Two no-flow boundaries are assigned along the shorter side walls. On the other two sides there are Dirichlet boundaries corresponding to a groundwater table elevation of $H = 7\,\mathrm{m}$ and $H = 0.9\,\mathrm{m}$ at the top and the bottom of the slope, respectively (Fig. 5). The initial groundwater table elevation is interpolated linearly between the two Dirichlet boundaries. The initial condi-

**Table 2.** Soil hydraulic parameters for the three test cases for Parflow and the coupled model

|  | $\theta_r$ [−] | $\theta_s$ [−] | $K_s$ [ms$^{-1}$] | $\alpha$ [m$^{-1}$] | $n$ [−] | $S_s$ [m$^{-1}$] | $S_y$[a] [−] |
|---|---|---|---|---|---|---|---|
| **1D flow** | | | | | | | |
| sand | 0.045 | 0.43 | $8.25 \cdot 10^{-5}$ | 14.5 | 2.68 | 0.0015 | 0.255 |
| loamy sand | 0.057 | 0.41 | $4.05 \cdot 10^{-5}$ | 12.4 | 2.28 | | |
| **2D flow** | | | | | | | |
| soil | 0.1 | 0.45 | $5.70 \cdot 10^{-4}$ | 1.65 | 2 | 0.0015 | 0.28 |
| **3D flow** | | | | | | | |
| unit 1 | 0.004 | 0.43 | $3 \cdot 10^{-6}$ | 3.6 | 1.6 | $10^{-6}$ | 0.28 |
| unit 2 | 0.004 | 0.41 | $7 \cdot 10^{-5}$ | 5.4 | 2.1 | | |
| unit 3 | 0.004 | 0.38 | $3 \cdot 10^{-7}$ | 2.7 | 1.6 | | |

a Only needed in the coupled model. Using the iterative coupling, this value is the initial value $S_y^0$.

tion for the 1D models is generated applying hydrostatic equilibrium and assigning a minimal initial pressure head of $-1.25$ m. Monthly varying rainfall (Fig. 6) is used as Neumann boundary condition for the land surface. The forcing is repeated for five cycles summing up to a total simulation time of five years. The soil parameters are listed in Table 2.

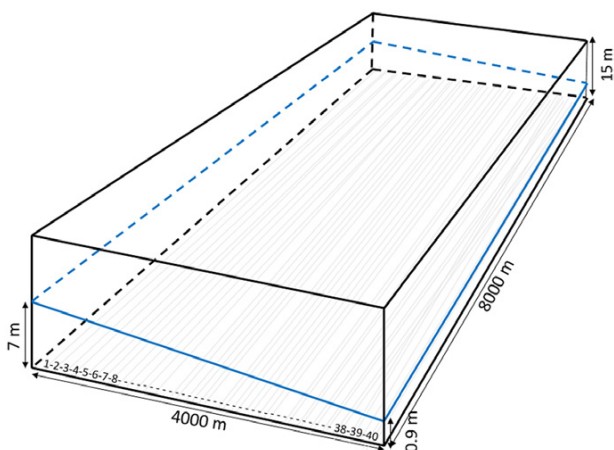

**Figure 5.** Model domain and initial water table position for the 2D flow model. (Beegum et al., 2018)

The 3D Parflow reference model uses a uniform grid with $80 \times 40 \times 150$ cells with grid size $\Delta x = \Delta y = 100$ m and $\Delta z = 0.1$ m. The spatial resolution in the coupled model is the same with a $80 \times 40$ grid for the groundwater model and 150 cells for the unsaturated zone models. The groundwater domain is divided into 40 zones all consisting of slices along the 8000 m side





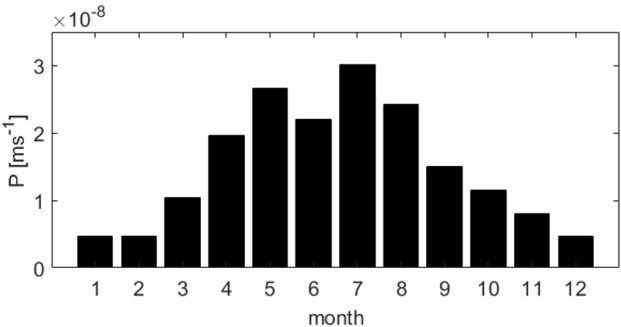

**Figure 6.** Net inflow used for the 2D flow model.

as shown in Fig. 5. With the flow problem being 2D this means that the entire domain is acutally covered by 1D models. The coupling and groundwater time step is $\Delta t_c = 1\,\mathrm{d}$ and the time step for the 1D models is again adaptive with a minimum value of $1\,\mathrm{s}$ and a maximum value of $1\,\mathrm{d}$. Parflow uses an adaptive time scheme as well, with a maximum value of $1\,\mathrm{h}$.

### 3.3 Test case 3: 3D flow

The first two test cases are rather simple. Therefore, a third test case is designed where the properties that are relevant for the coupling are closer to those of a model for a real hydrological system. The setup is similar to that of the second test case but with some modifications. The domain size is reduced to $800\,\mathrm{m} \times 400\,\mathrm{m} \times 15\,\mathrm{m}$, the slope is maintained. The lateral boundaries are the same, the flux boundary at the land surface now consists of three cycles of the precipitation shown in Fig. 6, followed by two no-flux years and then again two years using the original precipitation data. Thus, the total simulation time is seven years. With

these changes, the groundwater table is forced to eventually fall and then rise again. Apart from the more dynamic groundwater table, the influence of heterogeneities should be investigated with this test case. Therefore, instead of the homogeneous soil, three different soil units are distributed throughout the domain as depicted in Fig. 7. The corresponding parameter values are given in Table 2. In the coupled model, the parameters for the groundwater model are averaged arithmetically over the depth of the saturated zone in that grid cell. This averaging takes place at every time step adapting to the new position of the groundwater

table. The initial condition is the same as in the 2D flow case.

The horizontal spatial resolution is increased to $\Delta x = \Delta y = 10\,\mathrm{m}$. In the vertical direction a non-uniform grid is used with smaller grid sizes close to the surface and a total of $50$ cells. The same resolution is used for Parflow and for the coupled model. With the heterogeneous soil, each grid cell would need a 1D model to cover the entire domain. As this is computationally not feasible, the groundwater domain is again divided into $40$ zones but now spread evenly across the domain each covering an

area of $10 \times 8$ grid cells. The 1D models are placed at the center of each zone. Thus, also the influence of a simplified recharge (and specific yield) pattern, which is constant within each zone and calculated from the corresponding 1D model, can be investigated. The time step sizes are the same as in the 2D flow model.



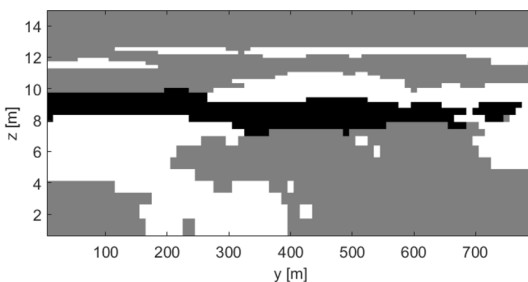

**Figure 7.** Distribution of soil units along the $800\,\mathrm{m}$ side at $x = 195\,\mathrm{m}$ in the 3D flow model. *grey:* unit 1, *white:* unit 2, *black:* unit 3.

## 3.4 Test case 4: sensitivity analysis

As already pointed out, for this test case the 2D and 3D flow test cases were combined. The domain size and the boundary
conditions as well as the initial condition are the same as in the 3rd test case. This was done to maintain the rising and falling
groundwater table. To get rid of spin up effects, the simulation time was increased, running four times the precipitation data used
for the 3D flow case, adding up to a total simulation time of 28 years. The soil structure was simplified to be homogeneous
within each model compartment, thus the groundwater model and the unsaturated zone models. For a heterogeneous soil
structure a global senstivity analysis would be neither feasible nor very meaningful. The parameters included in the sensitivity
analyis, along with the ranges of the uniform distribution functions $\rho$ they are sampled from, are listed in Table 3. Additionally,
the influence of the upper boundary condition is investigated by multiplying the precipitation data with a constant factor $r_{\mathrm{PET}}$.
The residual saturation $S_r = \theta_r/\theta_s$ and the specific storage $S_s$ are excluded from the analysis and set to 0.01 and 0.0015,
respectively.

**Table 3.** Parameter ranges for the parameters included in the sensitivity analysis. For the saturated hydraulic conductivity values for the
groundwater (GW) and the unsaturated zone (UZ) model as well as the van Genuchten $\alpha$ the normal logarithm is sampled.

|  | $K_{GW}$ [ms$^{-1}$] | $S_y$ [−] | $K_{UZ}$ [ms$^{-1}$] | $\alpha$ [m$^{-1}$] | $n$ [−] | $\theta_s$ [−] | $r_{\mathrm{PET}}$ [−] |
|---|---|---|---|---|---|---|---|
| min. | $5 \cdot 10^{-7}$ | 0.05 | $5 \cdot 10^{-7}$ | 0.5 | 1.5 | 0.05 | 0.75 |
| max. | $10^{-4}$ | 0.7 | $10^{-4}$ | 10 | 5 | 0.7 | 1.25 |

The horizontal spatial resolution is again $\Delta x = \Delta y = 10\,\mathrm{m}$, whereas the vertical resolution is $\Delta z = 0.1\,\mathrm{m}$ as in the 2D flow
case. As the sensitivity analysis requires a lot of model realisations, the number of unsaturated zone models and corresponding
zones is decreased to eight to save computation time. Similar to the 2nd test case they are placed evenly along the $800\,\mathrm{m}$ side
of the domain since the soil is homogeneous again. The time step sizes are the same as in the previously described test cases.





## 4 Results and Discussion

The coupling methods are compared in terms of accuracy and efficiency. The spatial and temporal distribution of the ground-
water table is used for the evaluation of the accuracy. The wall clock time serves as indicator for speed. The iterative coupling
uses a dynamic formulation of the specific yield. The resulting values are investigated in terms of plausibility. The groundwater
table position, the groundwater table fluctuations and the specific yield calculated from the iterative coupling (all taken at the
center of the domain) are used as observations $f(\boldsymbol{x})$ for the global sensitivity analysis. The resulting transient activity scores
(Eq. 13) of the parameters listed in Table 3 are compared for the two coupling methods.

### 4.1 Results 1D flow

Figure 8 shows the water table elevation over time for the single column reference run and the two coupling schemes. The
reference elevation $z = 0\,\mathrm{m}$ is located at $5\,\mathrm{m}$ below the surface. This is done to facilitate the comparison to the work by
Beegum et al. (2018). Both, the iterative and the non-iterative coupling scheme show a good match with the single column run.
The accuracy of the iterative approach is slightly higher, no difference to the single column run can be seen in Fig. 8. A visual
comparison indicates that the coupling applied by Beegum et al. (2018) yields a comparable accuracy.

The dynamic adaptation of the specific yield in the iterative coupling scheme is active only in the presence of lateral fluxes,
because otherwise due to the formulation of the recharge in Eq. 7 the groundwater table fluctutations in the two model com-
partments are already forced to be the same in the first iteration step. Therefore, in this test case the value stays constant at its
initial value and further investigation is not possible.

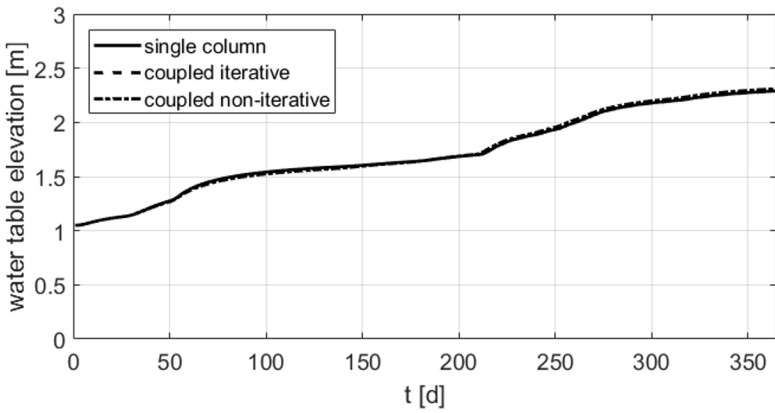

**Figure 8.** Groundwater table elevation over time for the 1D flow problem.

The wall clock times for the different methods are given in Table 4. No information about run times are given in Beegum
et al. (2018). Therefore, their results are not included in the speed comparison. All Matlab calculations are done on a local
desktop PC with an Intel Core i7 using 4 cores for the coupled model runs. The Parflow runs used for the other test cases are
performed on the JUWELS supercomputer at the Research Centre Jülich which provides 2567 compute nodes with Dual Intel



Xeon CPUs. 12 cores were used for the simulations. For the 1D flow problem, the iterative coupling takes roughly twice as
much wall clock time as the non-iterative coupling.

**Table 4.** Wall clock times for the different test cases and coupling schemes.

|  | control run | iterative coupling | non-iterative coupling |
|---|---|---|---|
| **1D flow** | 40 s (Matlab) | 82 s | 35 s |
| **2D flow** | 12960 s (Parflow) | 799 s | 346 s |
| **3D flow** | 18000 s (Parflow) | 4215 s | 625 s |

## 4.2    Results 2D flow

The temporal evolution of the groundwater table position at the center of the domain is shown in the upper plot of Fig. 9. While
both coupling approaches tend to overestimate the elevation of the groundwater table, increasing with time, the non-iterative
coupling shows a rather large offset compared to the Parflow run. This difference is also evident in the spatial plot of the
groundwater table in the lower part of Fig. 9. The overestimation of the iterative coupling scheme is also visible there, but is
clearly smaller. Besides, there is a shift in the shape of the groundwater table. The iterative coupling scheme shows the largest
difference to the Parflow model close to the boundary at $x = 4000\,\mathrm{m}$ which will be discussed further in Sec. 4.5. The largest
difference between the non-iterative coupling and the Parflow run occurs at around $x = 1000\,\mathrm{m}$. The results by Beegum et al.
(2018) have a similar accuracy and shape as the results of the iterative coupling approach.

When considering the non-iterative model, it is notable that initial time steps are an issue, as the model starts almost imme-
diately with a too high groundwater table. However, the difference also grows slowly over time. Both of these issues may be
related to the reference model essentially acting as a bucket without any plausible steady state solution (i.e. steady state for
the groundwater model would have groundwater tables far above the top of the domain). As such, an initial error in the model
can never be compensated and will also lead to feedback effects, such as an overestimation of the groundwater recharge due to
too short unsaturated zone model columns. In this respect, the 2D flow test case is of limited use when evaluating simplified
models for realistic setups.

The lateral fluxes in this setup cause a temporally and spatially changing value of the specific yield in the iterative model,
which can be seen in Fig. 10. However, the differences among the predefined zones (different lines) and also the temporal
fluctuations are rather small. All values are smaller than the proposed value of 0.28, although the difference is less than 0.03.
The spatial differences of the $S_y$ values indicate the dependence on the water table position whereas its temporal fluctuations
suggest a dependence on the fluctuations of the groundwater table which also show a repeating pattern due to the cyclic forcing.
Nevertheless, both effects are small here, which might be due to the homogeneity of the soil.

For this setup, again the non-iterative coupling is about two times faster than the iterative coupling (see Table 4). The run
time of the full 3D Parflow model is much higher.

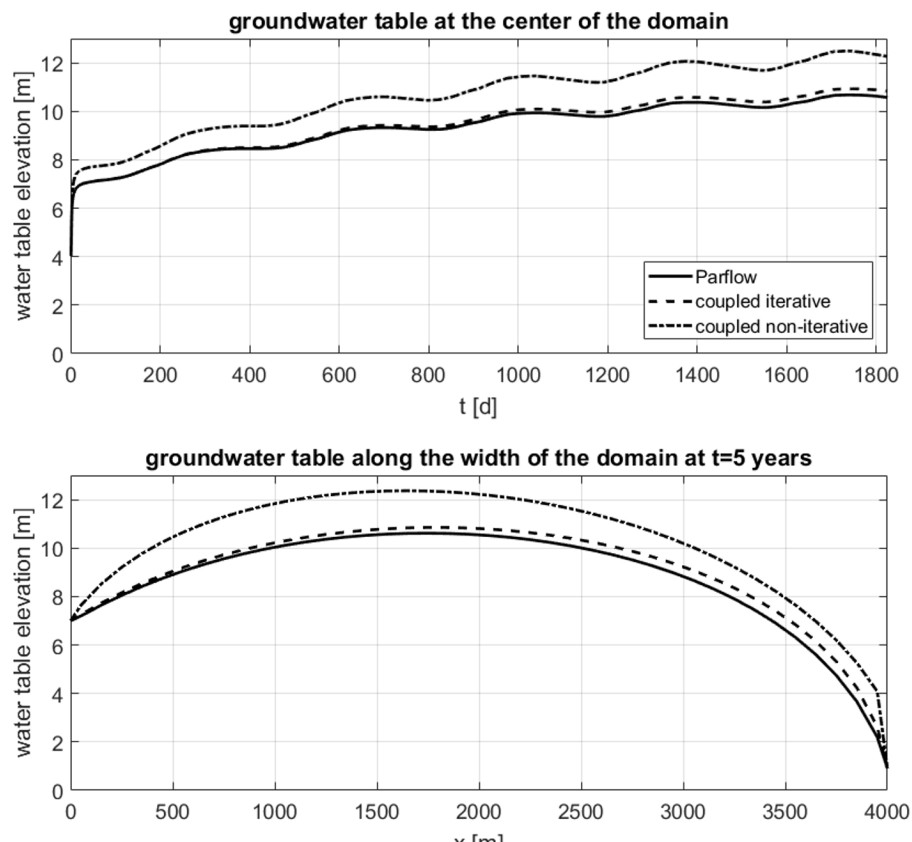

**Figure 9.** Results of the 2D flow problem. *Upper:* Groundwater table elevation over time in the center of the domain. *Lower:* Groundwater table along the width of the domain at the final time step.

### 4.3  Results 3D flow

In Fig. 11 the position of the groundwater table at the center of the domain is plotted over time (upper part). Both coupling schemes show a good agreement with the fully integrated 3D model. Similar behaviour is found at other positions in the domain. The non-iterative scheme has a slight delay when the groundwater table is rising more rapidly, e.g. around $t = 1000\,\text{d}$ and $t = 2250\,\text{d}$. The groundwater table along the width of the domain ($y = 400\,\text{m}$) at the end of the simulation is plotted in the lower part of Fig. 11. Again, both coupling approaches match the Parflow result very well with small differences between $x = 300\,\text{m}$ and $x = 370\,\text{m}$.

The heterogeneity of the soil causes more variability of the groundwater table throughout the domain. Figure 12 shows the difference between the groundwater table resulting from the coupled models and the full 3D model at the last time step in the entire domain. It can be seen that in the major part of the domain the differences are small. Areas with larger differences appear at similar locations for both coupling schemes showing the largest deviations of up to $\Delta H_{GW} = 1.5\,\text{m}$ along the $y = 800\,\text{m}$





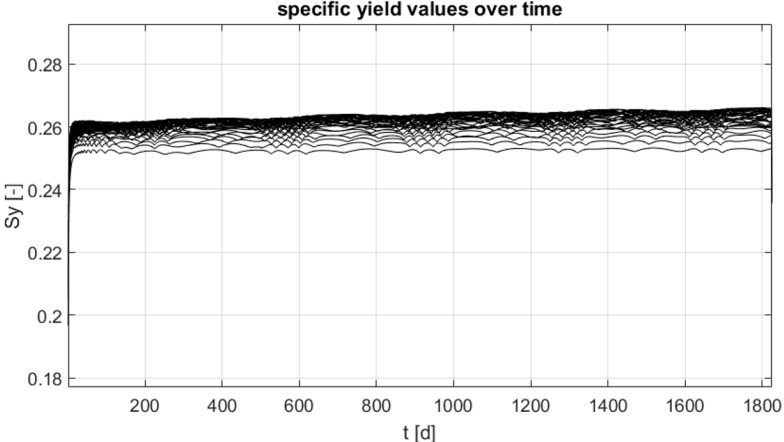

**Figure 10.** Specific yield values resulting from the iterative coupling for the 2D flow problem. Each line corresponds to a 1D model and zone of equal recharge and specific yield in the groundwater model.

boundary. Within these areas the differences between the iterative coupling and the Parflow model are smaller compared to the non-iterative coupling.

The heterogeneity also leads to a larger variability of the specific yield in the iteratively coupled model as can be seen in Fig. 13. There is a significant variation in space and time, with values ranging between approximately $0.01$ and $0.7$. Overall, the
specific yield values are decreasing when the groundwater table is rising and increasing when the groundwater table is falling (roughly between $t = 1100\,\mathrm{d}$ and $t = 2200\,\mathrm{d}$).

The ratio of the run times (Table 4) is different for this case, with the iterative coupling using almost seven times more wall clock time than the non-iterative coupling. While the Parflow model and the non-iteratively coupled model need less than twice the wall clock time than for the 2D test case, the wall clock time of the iteratively coupled model increases by a factor of five.
Still, both coupled models are notably faster than the fully integrated 3D model.

### 4.4 Results GSA

A global sensitivity analysis is performed to identify the most influential parameters of the two coupled models regarding the groundwater dynamics, i.e. the groundwater table elevation $H_{GW}$ and its daily fluctuation $\Delta H_{GW}$, both taken at the center of the domain. The resulting normalized transient activity scores $NAS$ are plotted in the lower two plots of Figs. 14 and 15. The
upper plots of Figs. 14 and 15 show the water table elevation and fluctuation of the iteratively coupled model averaged over all realisations, respectively. All model realisations in which flooding occurs are removed from the analysis as the model is not able to represent overland flow properly. Therefore, in the iteratively coupled model 316 out of initially 500 realisations are used and 395 in the non-iteratively coupled model.

Figure 14 shows similar trends and patterns of the activity scores for the two coupling schemes. In both cases, the saturated
hydraulic conductivity of the groundwater domain $K_{GW}$ is by far the most influential parameter, except for in the beginning.

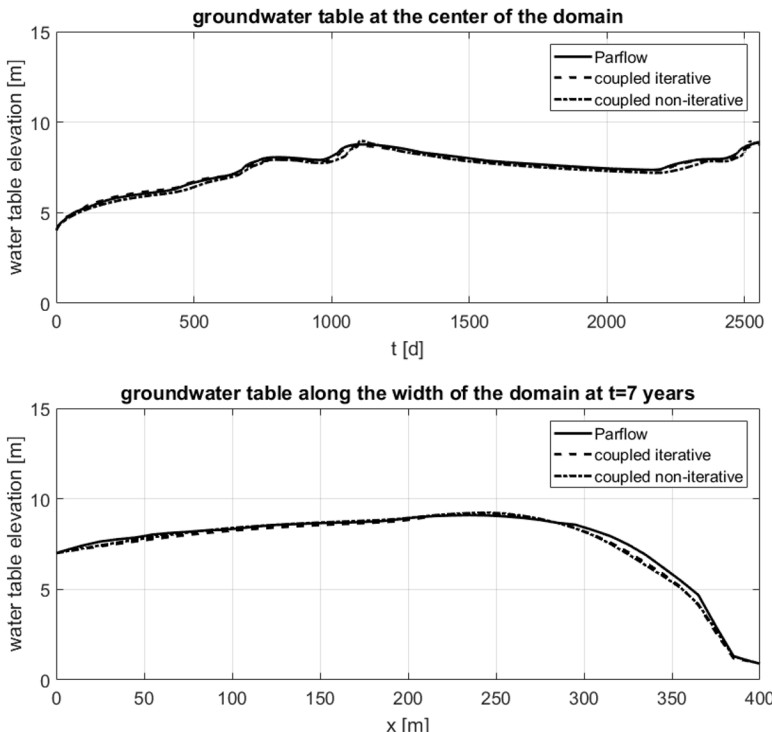

**Figure 11.** Results of the 3D flow problem. *Upper:* Groundwater table elevation over time in the center of the domain. *Lower:* Groundwater table along the width of the domain ($y = 400\,\mathrm{m}$) at the final time step.

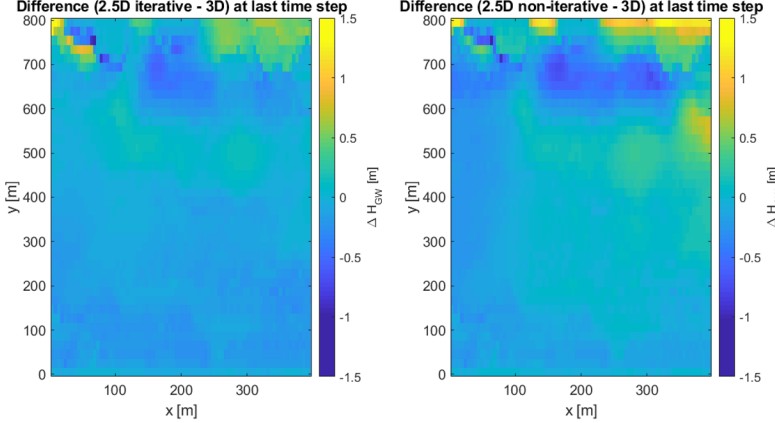

**Figure 12.** Difference of the groundwater table position compared to the full 3D run at the last time step. *Left:* iterative coupling. *Right:* non-iterative coupling.





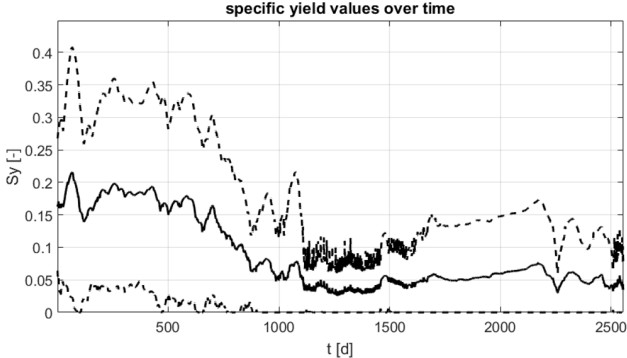

**Figure 13.** Specific yield values resulting from the iterative coupling for the 3D flow problem. The solid line is the spatial average, the dashed lines indicate ± one standard deviation.

There, the van Genuchten parameter $\alpha$ is dominant but its influence decreases with time. This is probably due to a spin up effect. When the model reaches the dynamic steady state (last two cycles), the influence of $\alpha$ has basically vanished. In the iteratively coupled model, porosity $\phi$ and the specific yield $S_y$ still have a small influence especially when the water table starts to rise again after a draining period. Note that the specific yield in the iteratively coupled model is not the value used for the non-iteratively coupled model defined in Table 3 but the value calculated by the model during the simulation. In the non-iteratively coupled model there is also a slight sensitivity to the specific yield. All other parameters play a minor role.

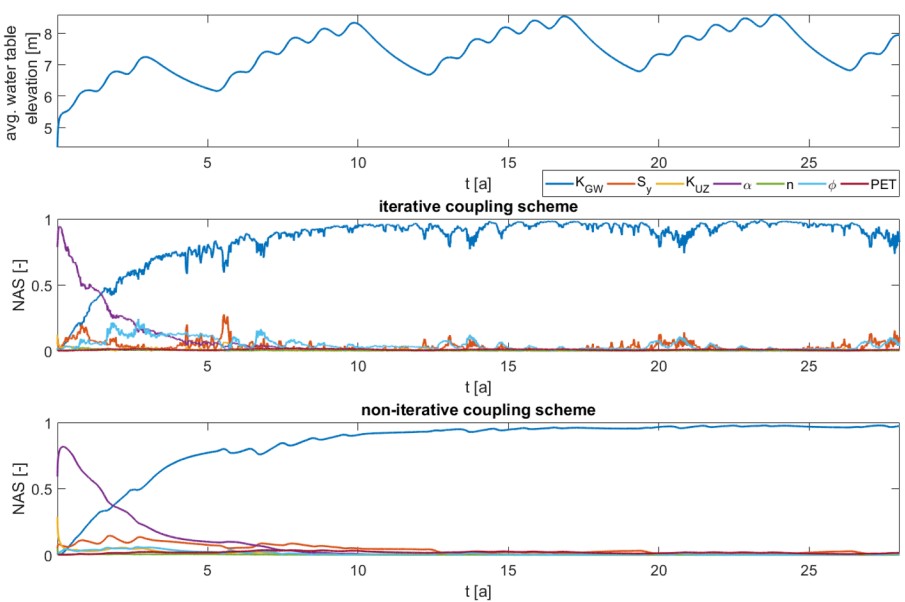

**Figure 14.** *Upper plot:* Averaged water table elevation of the iteratively coupled model. *Lower plots:* Transient normalized activity scores for the two coupling methods.





The acitivity scores concerning the water table fluctuations look different as can be seen in Fig. 15. Here, the most influential parameters are porosity $\phi$, the saturated hydraulic conductivity of the groundwater model $K_{GW}$ and the specific yield $S_y$. The dominance of these parameters is alternating. The saturated hydraulic conductivity $K_{GW}$ is the most important parameter during no rain periods. The fluctuations are then mainly caused by lateral fluxes, which depend strongly on this parameter. In the iteratively coupled model, when a high or low peak in $\Delta H_{GW}$ is reached, the recharge is the main driving force which is then mostly influenced by porosity. When looking at Eqs. 1 and 7, one sees that $S_y$ can be eliminated which explains why there is no influence of $S_y$ under these conditions. These are the extreme conditions and inbetween the specific yield has the largest influence. At these times, there is also an increase in the influence of $K_{UZ}$. As will be shown later this is because the calculated specific yield mainly depends on this parameter. The results for the non-iterative model are very similar except for that the scores of porosity and specific yield follow a slightly different temporal pattern and the influence of the remaining parameters is a bit larger, but still rather small.

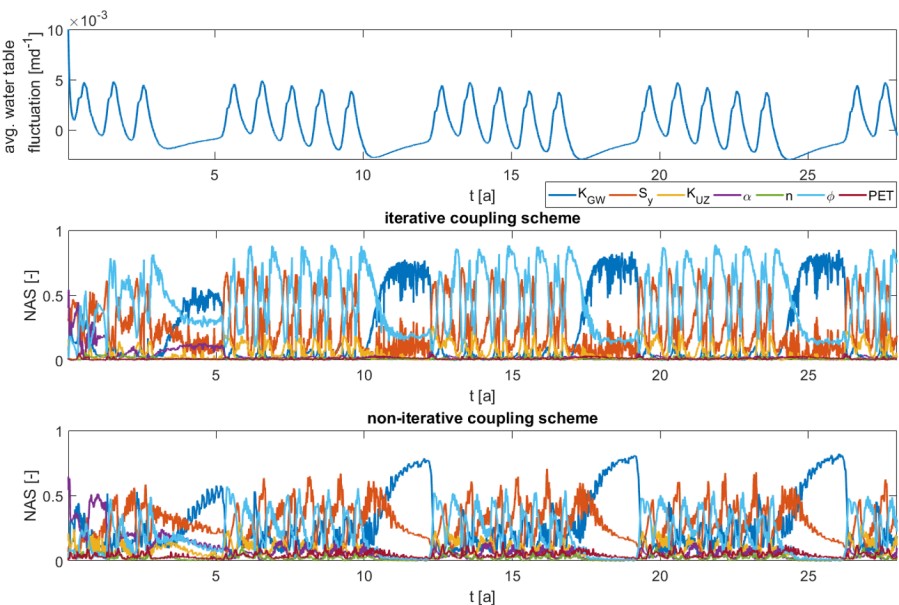

**Figure 15.** *Upper plot:* Averaged water table fluctuation of the iteratively coupled model. *Lower plots:* Transient normalized activity scores for the two coupling methods.

Thus, we see the importance of the specific yield for the groundwater table dynamics. With the iterative calculation of the specific yield a prior calibration of one of the most influential parameters is not needed. To see whether this calculated parameter behaves reasonably, a sensitivity analysis towards the calculated $S_y$ was conducted. The results are shown in Fig. 16. Note that the $S_y$ in the parameter list is now the initial value defined in Table 3 to which no sensitivity is expected. The average $S_y$ value shows some smaller fluctuations, but overall it converges to a value around $S_y = 0.17$, which is a plausible value. The activity scores are also on the whole constant in time with the two most influential parameters being porosity $\phi$ and the

saturated hydraulic conductivity of the unsaturated zone model $K_{UZ}$. These two parameters are followed by the van Genuchten

parameters $\alpha$ and $n$, whose influence is already much smaller. Besides, there is a small influence of the saturated hydraulic conductivity of the groundwater model $K_{GW}$. This means that the specific yield is mainly depending on the unsaturated zone parameters. This is reasonable as its intention is to represent the missing unsaturated zone in the groundwater model.

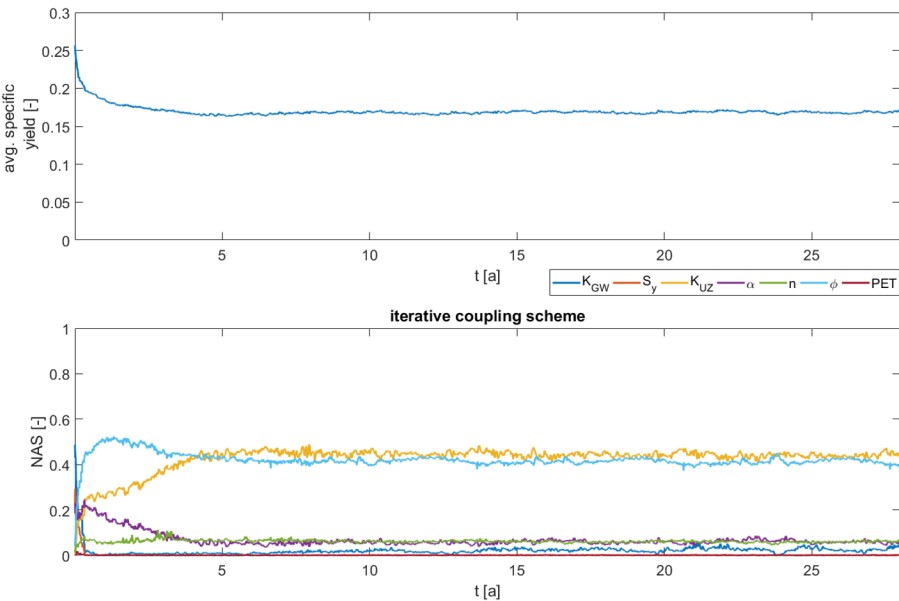

**Figure 16.** *Upper plot:* Averaged specific yield values of the iteratively coupled model. *Lower plot:* Transient normalized activity scores for the iterative coupling method.

## 4.5 Discussion

The above described numerical experiments show that both coupling methods presented in this work are well suitable to

substitute a full 3D model under certain conditions. They can correctly capture the dynamics of the groundwater table in presence of lateral groundwater flow and soil heterogeneities. Their accuracy is comparable to that of other coupling approaches (Beegum et al., 2018), but they are easy to implement, fast and in the case of the iterative model even consistent.

In the second test case, the non-iterative method overestimates the groundwater table elevation notably, probably rooting in an overestimation of the recharge flux. In the other test cases it also leads to slightly larger deviations from the 3D model

than the iterative coupling scheme. On the other hand, the higher accuracy of the iterative approach comes with a higher computational demand. Especially the heterogeneities in the third test case cause a large increase in run time compared to the non-iterative approach while the results are of comparable quality.

A higher computational efficiency could be achieved by using less 1D models. On the contrary, using more models could help decreasing the discrepancies in the less accurate areas close to the no-flow boundary at $y = 800\,\mathrm{m}$ which are most likely





caused by the soil heterogeneities and the simplified recharge and specific yield pattern due to the zonation. As this is a general

issue for these kind of models and does not relate to the presented coupling strategies themselves, we do not investigate it

further. Another general problem of the coupled model can be seen in the second test case. Because of the low groundwater

table and the large gradient at the lower Dirichlet boundary ($x = 400\,\text{m}$), water is also leaving the domain laterally via the

unsaturated zone. This cannot be captured by the coupled model. The water thus has to first move vertically through the 1D

model before it can leave the domain through the groundwater. This leads to a pile-up of water and therefore an overestimation

of the groundwater table along that boundary. This can also be seen in Beegum et al. (2018) which supports the assumption that

it is a general issue. In the 3D flow case the gradient at this boundary is smaller causing smaller lateral fluxes in the groundwater

and thus also in the unsaturated part. Therefore the results of the coupled model are on average more accurate even though this

test case is more complex than the 2D flow case.

The global senstivity analysis shows that the two coupled models behave reasonably and consistently among each other.

The groundwater table position $H_{GW}$ is mainly depending on the saturated hydraulic conductivity in the groundwater model

$K_{GW}$. This was to be expected as it is the main parameter in the steady state formulation of Eq. 1. The initially high and

then decreasing influence of the van Genuchten parameter $\alpha$ indicates larger fluxes in the unsaturated zone at the beginning

of the simulation. This is most likely due to the rather artificial initial condition in the unsaturated zone which is constantly

$h_{UZ} = -1.25\,\text{m}$ at $\geq 1.25\,\text{m}$ above the groundwater table and needs to be corrected during the spin-up phase. The temporal

fluctuations of the groundwater table are influenced by more parameters, namely the saturated hydraulic conductivity in the

groundwater model $K_{GW}$, the specific yield $S_y$ and porosity $\phi$. Which parameter is dominating depends on the current flow

conditions.

The dynamic formulation of the specific yield in the iterative coupling approach allows the groundwater model and the

unsaturated zone model to respond comformably to the given fluxes. So far this has been a challenge when using an overlapping

coupling approach (Zeng et al., 2019). The specific yield values obtained in these experiments confirm the assumption that it

is depending on the saturation profile, the fluctuations of the groundwater table and the soil properties. However, these values

should be interpreted with caution. This becomes evident when looking into the specific yield values of the 3D flow problem.

Physically, a value larger than porosity is not possible. Here, the task of the specific yield is to represent the unsaturated zone

in the groundwater model. A value that exceeds porosity means that a part of the water entering the cell laterally (if $Q > 0$)

does not add to the groundwater but flows upward into the unsaturated zone. In case of a negative lateral flux, water would

then be draining from the unsaturated zone. So actually this part is substracted from/added to the recharge. If this effect can be

accounted for correctly in the quantification of the recharge, the specific yield stays in its physically meaningful range. This

is not the case in this model as we cannot calculate this effect properly and we therefore keep $\Delta H$ due to recharge fixed (see

Eq. 7).

In the end, the specific yield is not a physical quantity but a model parameter. Therefore, it should be treated as such and

fitted for each application. Calculating it within the iteration substitutes its calibration and is therefore an advantage over other

methods. The sensitivity analysis for the parameters' influence on the calculated specific yield shows that it behaves reasonably

even though it can take unphysical values to compensate other effects. It depends on the parameters of the unsaturated zone





models, especially on porosity $\phi$ and the saturated hydraulic conductivity $K_{UZ}$. The aim of the specific yield is to represent the missing unsaturated zone in the groundwater model, therefore a strong dependency on the unsaturated zone model's parameters is plausible. So even though it is a model parameter, its values are meaningful.

## 5   Conclusions

In this work, two different ways of coupling a 2D groundwater model with multiple 1D unsaturated zone models are presented
and compared in terms of accuracy and efficiency. The first approach uses a moving boundary formulation and couples the two domains non-iteratively. In the second approach they are coupled iteratively and the 1D models overlap with the groundwater model throughout the entire saturated part. A recalculation of the specific yield value during the iteration is applied to get to the same water table position in both model compartments.

The results and run times of the two coupling methods are compared to those of fully integrated model runs. While the
iterative coupling shows a higher accuracy, the non-iterative approach is faster in which both models are significantly faster than the fully integrated model. Besides that, the iterative model appears to be more robust having a decent accuracy for all applied test cases, which is not the case for the non-iterative model, and can keep the consistency of the two model compartments.

A global sensitivity analysis revealed that the groundwater table position and its dynamics depend mainly on three parameters: the saturated hydraulic conductivity of the groundwater model, the specific yield and porosity. The specific yield is,
however, hard to quantify especially because it is usually not constant over time, fluctuations can be large in the presence of soil heterogeneities. In the iteratively coupled model the specific yield is calculated by the model itself during the iterations accounting for this variability. It depends mainly on one additional parameter which is the saturated hydraulic conductivity of the unsaturated zone model. This is an advantage of this method over other existing coupling strategies as this parameter is easier to be quantified and constant over time.

In general, both models qualify for substituting a fully integrated model when computational efficiency is more important than highly accurate results. The iterative approach is more robust and accurate and is recommended for most applications, while the non-iterative approach is better when speed is essential. Erdal et al. (2019) used a version of the here described non-iteratively coupled model successfully to reduce the compute time needed for model spin-up. Another application where such surrogate models can play an important role is data assimilation. Data assimilation often requires an ensemble of model
realisations which comes with a strong computational burden. How well the here presented 2.5D model can replace a fully integrated model in this context is part of ongoing research.

*Code and data availability.*   The MATLAB code to run the coupled model using the two presented coupling strategies and to visualize the output is available at http://doi.org/10.5281/zenodo.4422532. This code also includes the input and reference data for the three used test cases.



*Author contributions.* Simulations and code development were performed by DE and NB. IN acquired the funding. All authors contributed to the design of the experiments, the analysis of the results and writing the paper.

*Competing interests.* IN is a member of the editorial board of the journal.

*Acknowledgements.* This research is funded by the German Science Foundation (DFG) in the framework of research unit FOR 2131 under NE 824/12-1, and the Collaborative Research Center CAMPOS (CRC 1253 CAMPOS - Catchments as Reactors). Computing time has been
provided by the Jülich Supercomputing Centre (http://www.fz-juelich.de)





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
