# Peer review of "Coupling saturated and unsaturated flow: comparing the iterative and the non-iterative approach"

_Hydrology and Earth System Sciences, 2021_

## Referee Comment (RC2)

**Review of "Coupling saturated and unsaturated flow: comparing the iterative and the non-iterative approach",**
**by N. Brandhorst, D. Erdal and I. Neuweiler**

**Submitted for publication in Hydrology and Earth System Sciences**
(https://doi.org/10.5194/hess-2021-15 - Preprint. Discussion started: 20 January 2021)

**Reviewer: L. Orgogozo**

The manuscript presents two methodologies of coupling for dimensionnally heterogeneous modelling of subsurface flow in the unsaturated zone and in the saturated zone. The interest of such an approach is that important diminutions of computation times may be obtained compared to fully 3D modeling approaches. The main drawback is that accuracies of the simulations are dommaged, still in comparison with 3D modelling, and more or less along the considered cases and coupling methodologies. Given the very large computation times that may be encountered in fully mechanistic hydrological modeling at the wtareshed scale, this problem is of great interest for the community of hydrological modeling. The manuscript contains an important material in terms of numerical results and provides relevant hints to compare the two coupling stategies under concern. Nevertheless the presentation of the considered theories and numerical experiments lack of rigor, and the writting of the manuscript is not clear enough. In some places additionnal computations may even be needed. Thus I think it should be thoroughfully reworks prior to publication. I recommend to reject the paper in its present form, and to encourage the authors to resubmit after having completing and improving it.

**General comments:**

- The considered equations should be defined more rigourously and rewritten. For instance, the double time derivative term in equation (2) is a non-standard formulation of Richards equation (see for instance Gottardi and Venutelli, 1993). I guess that considerations related to the order on magnitudes of those two time derivatives may be used to justify the adopted formulation, but it should be explicited. Moreover, the use of the same notation $S_y$ for the specific yield in equation (1), which has classically a clear and well identified physical meaning (drainage porosity), and for the iteratively computed, time variable fitting parameter used in the iterative method to handle recharge fluxes from the unsaturated zone is confusing. I think that the latter one should be expressed as the sum of the true specific yield $S_y$ and a new additional term used for the purpose of the coupling between the saturated zone and the non-saturated zone. This would not imply new computation, but simply to rewrite some equations and rescale some results. I think that the added value of such more accurate notations in terms of clarity and of ease of physical interpretation would be important.

- Convergence studies for mesh refinement and time stepping strategy are not evocated as it should be the case in any study producing Computational Fluid Dynamics results. In some places it may imper the possibility to understand the comparative behaviours of the proposed test cases. For instance if we consider the comparison of accuracies of test 2 and test 3, in the present form of the manuscript it is impossible to say what comes from the differences of meshes and what comes from the different physics under concern (e.g.: homogeneous versus heterogeneous soil).

- To the knowledge of the reviewer, an important example of hydrological model that couples dimensionnaly heterogeneous descriptions of flow in the saturated zone and in the unsaturated zone is MIKE-SHE (e.g.: Graham and Butts, 2005), which is for instance included in recent international benchmarking efforts for physically based hydrological modeling (e.g.: Kollet *et al.*, 2016). The fact that works related to MIKE-SHE do not appear in the references of the manuscript make me think that the bibliographical survey on which the presentation of the background of the study is done should be consolidated.

**Specific comments:**

- l 136-137: "a Neumann boundary representing net flux from precipitation and evapotranspiration" : with the source/sink term of the equation (2), it is possible to represent actual evapotranspiration distributed in time and space according to water avalaibility in the soil (see for instance Orgogozo et al., 2019) ; please discuss the limitation associated with an a priori estimation of the actual evapotranspiration directly embedded in the Neumann boundary.

- l 151 : "collect the computed recharge (i.e. flux leaving over the bottom boundary) and interpolate the 2D map of groundwater recharge." : You mean collect all the computed regarges for all time steps of the 1D Richards model since the previous time step of coupling ? Should be clarified.

- l 157 : "Add (or subtract) a ratio r of this water to the recharge computed in the next time step." you mean the next time step of coupling ? Should be explicited.

- l 159 – 171 : The proposed way of chosing the ratio r is difficult to accept. In case of water table elevation, the ratio r could be fitted to keep unchanged across the mesh resizing process the total amount of water contained in the part of the domain that stays unsaturated, while in case of water table lowering a 'field capacity' water saturation could be prescribed to the cells that experienced desaturation in order to compute a total water amount to be distributed in the new 1D mesh, with an associated proper r ratio? Here the formulae proposed for the computation of the ratio r seems somewhat arbitrary. For instance the point (1) "the groundwater table rise or fall is also effected by lateral flows" is already taken into account in the 2D groundwater model. Besides, "the unsaturated zone is really compacted by a rise of groundwater levels" does not sound physical at all.

- l 183 : "(v) the iteration counter" : with which loop is related this iteration counter is unclear at this point (it could be for instance with the time stepping of the 1D Richards equation or with the coupling time steping)? Although it becomes clear afterward, it should be explicited here, at the first occurence of (v).

- l 203 eq (10) (see the first general comment): According to the basic derivation of the diffusivity equation for unconfined aquifer, the specific yield is equal to the drainage porosity of the considered porous medium – although it seems that it might be different for more elaborated derivations, according to the literature cited by the authors. What is the physical interpretation of the variations the specific yield computed by eq (10)? Is there a theoritical reason why the iterating on the values of the specific yield field in the aquifer should lead to convergence? In case this is a purely empirical methodology, are there cases for which divergence may occur? Other questions : the value of the 'physical' $S_y$ parameters that appears in the equation (1) is only the seed of the iterative process at the first time step of simulation, and do not appear directly anymore in the course of the simulation for the evaluation of $S_y^v$, right? That is what I understand from table 2 for instance. It should be clarified here.

- l 221 – 224 : "The source/sink terms $q_{lat,i}$ have an effect on the recharge ('R(Q lat = 0) ≠ R(Q lat ≠ 0)'), which due to the nonlinearity of Richards equation cannot be quantified." However at step (4) (l 216 – 217), an updated $R^v$ is computed that takes into account the $q_{lat,i}{}^v$? I don't understand.

- l 229 – part 2.3 Activity score: Difficult to follow. Lack of explanations and of references. There is also a problem of structure: since 'The parameters and the model output f are defined in Sections 3.4 and 4.4, respectively.', this part 2.3 is not possible to understand by itself at this point of the reading. This part should be reworked so that the reader may understand why it is interesting to use the activity scores for the sensitivity study, and on what motivated the choices of the parameters x and the output f(x). By the way, in table 3 part 3.4, the variable $K_{GW}$ and $K_{UZ}$ seems not to be defined in the manuscript ? And why chosing $S_y$ as a parameter of the sensitivity study while it is subjected to iterative evolution of its value along computation in the iterative method (l 218 – table 2)?

- l 267 : '3.1 Test case 1: 1D flow' lack of a figure that presents the geometry, the boundary conditions and the meshes for each models.

- Figure 4 : Wrong title for y-axis (Precipitation-PET, not just PET)

- l 276 : Precise which 1D model (pure Richards I suppose ? )

- l 281 : "The groundwater domain is divided into a 2 × 2 grid. Each groundwater cell is assigned a 1D model." Then the groundwater model is 2D with only 2 cells in each direction ? I don't understand.

- l 285 : "Since there is no variability along the 8000 m side, flow is effectively 2D in this test case." Then it is useless and misleading to present it as a 3D computation ; the figures and the discussions should be reshape for presenting directly the test case as a 2D one. The comparison of computation times is also questionnable : to deal with a 2D case in 3D increase tremendously (and artificially) the computation time with a fully mechanistic 3D model. Here some additionnal simulations (dealing with the 2D problem in 2D) are needed for making the comparison of computation times.

- l 290 : "[...] assigning a minimal initial pressure head of −1.25 m" ; you mean that -1.25 m is the pressure head at the top of the domain ? Please clarify.

- l 291 : "Monthly varying rainfall (Fig. 6) is used as Neumann boundary condition for the land surface". More precision about these data would be useful – e.g.: are they synthetic ? Of which type of climate are they representative ?

- l 292 : Table 2 is not timely introduced ; since it contains information for the 3 test cases, it should be placed either in the beginning or at the end of the presentation of the considered test cases, but not at the middle.

- l 293 – 294 : "grid size $\Delta x = \Delta y = 100$ m and $\Delta z = 0.1$ m.". It makes a form factor of $10^3$ ... Any convergence study done for the mesh refinement?

- l 296 : "With the flow problem being 2D this means that the entire domain is acutally covered by 1D models." Nevertheless as far as I understood the proposed methodologies it would be exactly the same if the case was a 3D one? And I don't understand to which extent a 1D approximation for a 2D problem would be essentially more "acute"  than a 1D approximation for a 3D problem?

- l 306-307 : "three different soil units are distributed throughout the domain as depicted in Fig. 7." More information is needed here. Is this distribution synthetic? How has it been acquired / built ? Of which type of soil (sand, loam, clay ...) each unit is representative ?

- l 308 : "averaged arithmetically" Any tests for the use of harmonic or geometric mean instead of arithmetic mean?

- l 309 : " In the vertical direction a non-uniform grid is used with smaller grid sizes close to the surface and a total of 50 cells." Please provide the minimum and maximum sizes.

- l 315 : "The 1D models are placed at the center of each zone." How are laterally averaged the porous medium properties in each 1D models covering 10*8 cells laterally?

- l 327 : "The residual saturation $S_r = \theta_r / \theta_s$ and the specific storage $S_s$ are excluded from the analysis and set to 0.01 and 0.0015, respectively." Why have they been excluded ? To be justified, or at least discussed.

- Table 3 : The parameters $K_{GW}$ and $K_{UZ}$ are appearing in the manuscript for the first time in this table. The notations used in table 3 and those used in the equations (especially (1) and (2)) should be the same, of at least explicitly related.

- l 329 -330 : "The horizontal spatial resolution is again $\Delta x = \Delta y = 10$ m, whereas the vertical resolution is $\Delta z = 0.1$ m as in the 2D flow case." Once again a convergence study must have been done to justify the use of this mesh with a form factor of $10^2$.

- l 332 : "The time step sizes are the same as in the previously described test cases." Any convergence study for justifying the use of the proposed time stepping policy?

- l 345 : "A visual comparison indicates that the coupling applied by Beegum et al. (2018) yields a comparable accuracy." Why not plotting the results of Beegum in Figure 8?

- l 356 Table 4: This table contains information for all test cases and then it is not at the right place, being presented in a part specific to test case 1. Besides, since in test case 1 there is no lateral flux and thus no iteration in the iterative methods, I wonder why the iterative method has a wall time twice time more long than non-iterative method, while this later one include an addtionel step of remeshing? To be discussed.

- l 363-364 : "The results by Beegum et al. (2018) have a similar accuracy and shape as the results of the iterative coupling approach." Why not plotting them in Figure 9?

- l 365 : "When considering the non-iterative model, it is notable that initial time steps are an issue [...]" Any convergence study on time step ? What happens if smaller time steps are used?

- l 366-368 : "Both of these issues may be related to the reference model essentially acting as a bucket without any plausible steady state solution (i.e. steady state for the groundwater model would have groundwater tables far above the top of the domain)." Then why not chosing lower values of precipitation , so that a steady state may be reached?

- l 374 : "All values [of $S_y^0$] are smaller than the proposed value of 0.28, although the difference is less than 0.03." How the proposed value of 0.28 has been choosen? Are there correlations between the $S_y$ values and the state of the groundwaters (e.g.: water table altitude, lateral fluxes intensity)?

- l 379 : "Both coupling schemes show a good agreement with the fully integrated 3D model." It is hard to understand why the matching between the fully 3D computation and the 2.5D ones is better here for this 3D heterogeneous test case than in the 2D homogeneous test case 2. I noted that in test case 3 a finer mesh is used than in test case 2. May be that convergence issues are at stake?

- l 389-391 : "Areas with larger differences appear at similar locations for both coupling schemes showing the largest deviations of up to ΔH GW = 1.5 m along the y = 800 m boundary". Why are they such discrepancies, and why there? These points should be discussed here.

- l 394-396 : "Overall, the specific yield values are decreasing when the groundwater table is rising and increasing when the groundwater table is falling (roughly between t = 1100 d and t = 2200 d)." Once more, a careful discussion of the physical meaning of $S_y$ and its variation is needed.

- l 414 : "Note that the specific yield in the iteratively coupled model is not the value used for the non-iteratively coupled model defined in Table 3 but the value calculated by the model during the simulation." I don't understand how it is possible to make a sensitivity analysis on a parameter that is not constant and specified prior to computation, but time-variable, calculated along computation?

- l 417 : "Acitivity"

- l 421-422 : "When looking at Eqs. 1 and 7, one sees that $S_y$ can be eliminated which explains why there is no influence of $S_y$ under these conditions." You mean that dh/dt = 0 at extremas ? To be clarified.

- l 431-432 : "The average $S_y$ value shows some smaller fluctuations, but overall it converges to a value around $S_y$ = 0.17, which is a plausible value.". This is a too short discussion of the value of this key parameter that controls the exchanges between the saturated zone and the unsaturated zone in the iterative method. It should be interpreted physically. It seems to potentially encompasses a non clearly identified list of physical phenomena.

- l 436-437 : "This means that the specific yield is mainly depending on the unsaturated zone parameters. This is reasonable as its intention is to represent the missing unsaturated zone in the groundwater model." Somewhat strange. According to the basic derivation of the diffusivity equation for unconfined aquifers, the specific yield should be a property of the saturated zone (drainage porosity). So may be that if this parameters depends mainly on the properties of the unsaturated zone, it means that it is not, or not only, a specific yield (see the first general comment)?

- l 442 : "in the case of the iterative model even consistent." I am not sure of what you want to say, please be more specific.

- l 448 : "On the contrary, using more models could help decreasing the discrepancies in the less accurate areas close to the no-flow boundary at y = 800 m which are most likely caused by the soil heterogeneities and the simplified recharge and specific yield pattern due to the zonation." These discrepancies are important (~1,5m), and their causes must be carefully assessed. Additional numerical

experiment with lower and stronger soil heterogeneities or various zonation startegies could help to ensure that the proposed diagnostic is correct. From my point of view stating that "As this is a general issue for these kind of models and does not relate to the presented coupling strategies themselves, we do not investigate it further." is not sufficient, at least without any bibliographical references as it is at present.

- l 458-459 : "Therefore the results of the coupled model are on average more accurate even though this test case is more complex than the 2D flow case." Meshes also are different, and without proper convergence studies the impact of this point may not be assessed. The convergence studies must be done, and used for consolidating the discussions.

- l 464 : "is constantly $h_{UZ} = -1.25$ m at $\geq 1.25$ m above the groundwater table" This should be made clear sooner (see teh comment on l 290).

- l 467 : "Which parameter is dominating depends on the current flow conditions." This should be discussed in more details.

- l 470 : "comformably" is not specific/quantitative enough.

- l 478-480 : "This is not the case in this model as we cannot calculate this effect properly and we therefore keep $\Delta H$ due to recharge fixed (see Eq. 7)." However in equations (7), (8) and (10), it is clear that there is an iterative procedure that involves $\Delta H_{UZ}^v$ and $\Delta H_{GW}^v$ that evolve at each iteration $v$? I don't understand.

- l 481 : "In the end, the specific yield is not a physical quantity but a model parameter.". This statement seems too general ; while it is clearly the case in the proposed modeling approach, it is not the case in all formulation of the diffusivity equation in unconfined aquifers. Overal all this paragraph should be rewritten to better discuss the meaning of the concepts that are specific to the proposed methodology with a wording that should not rise ambiguities between these concepts and previously existing concepts. For instance:

- l 485-486 : "The aim of the specific yield is to represent the missing unsaturated zone in the groundwater model" You are talking about what you called a specific yield in your model. I think that it should have another name that 'specific yield', this latter word designing a concept that do have physical meaning and that is related to the properties (drainage porosity) within the saturated zone in the basic form of the diffusivity equation for unconfined aquifers (see the first general comment).

---

## Author Comment (AC1)

Review of Brandthorst et al. 'Coupling saturated and unsaturated flow: comparing the iterative and the non-iterative approach'

We thank the reviewer for the effort and time to revise our manuscript and in the following want to respond in detail to the constructive comments he provided.

Major comments.

The contribution of the paper is purely computational. This can be seen from the test cases, all of which are highly artificial. I do not consider this a problem. The paper is focused, does not claim more than it delivers, and what it delivers is relevant for the HESS readership and substantial. The paper is well organized and generally clear. In the minor comments below I indicate where I was not sure if I understood everything.

We thank the reviewer for this rating. We agree that the test cases are entirely artificial. The aim was to apply and compare the methods under fully 'controllable' conditions. Evaluating the performance of the models in a more realistic setting or even a real-world case would be interesting work for the future.

Minor comments.

The conclusion in the abstract that the more elaborate iterative coupling is better and more robust but slower than the non-iterative coupling is a bit underwhelming. Perhaps being more specific would make it more informative.

We propose to reformulate this sentence to: "The non-iterative approach is faster but does not yield a good accuracy for the model parameters in all applied test cases whereas the iterative one gives good results in all cases. Which strategy is applied depends on the requirements: Computational speed vs. model accuracy."

The overview in the Introduction of the various coupled unsaturated-saturated models, their coupling strategies and the role of the specific yield is insightful and thorough.

Thanks.

l. 124- 131. If you assign the same 1D unsaturated zone columns to multiple grid cells of the underlying 2D groundwater model, I believe you are making the implicit assumption that the weather is uniform over the 2D extent of the groundwater level, or at least over the regions assigned to each of the 1D columns. If this is indeed the case, it may be good to mention this.

Yes, this is the case. We will change l.127-129 to "The atmospheric forcing of the unsaturated zone model as well as the recharge that is calculated from the unsaturated zone model and passed to the groundwater model are assumed to be constant within each zone. "

l. 156-173. If I understand correctly, if the resizing of an unsaturated zone column shortens the column, you have to add the storage in the part of the column that is cut off to the amount of water in the saturated zone to conserve mass during the resizing operation. When you increase the size of an unsaturated column, some water is transferred from the saturated to the unsaturated zone. I believe this is done in steps 4 and 5. In that case, I believe the ratio $r$ should normally be equal to 1 to ensure mass conservation. You give several reasons to deviate from this. Equation (6) gives the expression for $r$ you used, capped at 1. If you consider the entire system, does capping $r$ at a maximum of 1 whilst allowing values smaller than 1 not necessarily lead to mass losses when resizing all unsaturated columns?

Incidentally, for this reason I do not understand why Erdal et al. (2019) set r to 0 instead of 1, but I am not reviewing that paper here. But it seems to indicate that I missed something here, unless a shortening/lengthening of an unsaturated zone column led to a corresponding increase/decrease in its volumetric water content to keep the total amount of water present in the column unchanged.

Indeed, the r=0 assumption is questioned by us as well, and that is why the new ratio is used. One needs to consider that the models are not strongly coupled. Setting the ratio to 1 causes a lot of numerical problems following the one-step-behind adding of the lost water. An example is oscillating groundwater tables with increasing magnitude. We therefore settled for the rather ad hoc solution presented in the manuscript. As the capping occurs both on elongating and shrinking of the unsaturated zone, it will have both a positive and a negative effect on the total water in the system, so the comment of the reviewer is not quite right. We see that this was not well explained in the original submission and it will be improved in the revised version. In particular, we will write explicitly that a factor r=1 would be needed to keep strict mass conservation.

Perhaps it is possible to have a 'before and after' figure, table, or water balance for the saturated and unsaturated domain (1 column only) in which you track what amounts of water go where during resizing.

While this is in principle an attractive idea, it should be noted that the non-iterative model, which, as the reviewer also points out, is taken from a previous publication on speeding up model spin-up (Erdal et al 2019), in this work serves mainly as a comparison to the iterative one. It is a conceptually quite ad hoc, but very fast model. Hence, we do not wish to spend more work on the inferior model, but rather focus on the much better performing iterative coupling, and we hope the reviewer can understand our point of view.

l. 158, 221. What is point i? Do you mean point 1?

Yes, it should be point 1. This is an artifact from the previous template using roman numerals.

Figure 2. Perhaps use a different color for the horizontal line denoting the phreatic level. (Minor point, please don't bother if this takes too much time.)

This is a good suggestion. We will change it.

l. 203. Equation (10) forces the specific yield to conform to the simultaneous gain/loss of water and the resulting change in in the groundwater level. Essentially, the specific yield is no longer a model variable but the ratio of the calculated water gain per calculated groundwater level change. The water level change is calculated using the 1D Richards' equation, which does not have the specific yield as a parameter. You thereby to a certain extent impose the unsaturated zone model upon the groundwater model and adapt one groundwater model parameter (the specific yield) to match both models. Interesting approach.

Yes, this is exactly what we do and what to our knowledge has not been tested before. We think that the approach is well explained by the reviewer and will explain it this way in the revised manuscript.

l. 226. 'constant' or 'spatially uniform'?

We mean "spatially uniform". We will change this.

Figure 3. I was expecting a yes/no decision at the closure criterion, looping back to the start of the iteration in case of 'no'.

Right, this is missing. We will add it.

l. 271-272. You state that you have hydrostatic equilibrium as the initial condition (zero flow), but in the next sentence state that you state that in most of the profile, the matric potential is vertically uniform, which amounts to a profile with unit gradient flow.

We agree that the way it is written is misleading, we will change the two sentences to: "The initial condition assumes a hydrostatic pressure profile in the lower 6.5m of the soil corresponding to an initial position of the groundwater table at 3.95m below the surface. In the upper 3.5m of the soil unit gradient flow is imposed by setting the pressure head to -0.283m. "

l. 281-282: repeats l. 276-277.

Not exactly. L.281 – 282 refer to the 1D reference model and l .276-277 to the coupled model. We will make that clearer.

l. 325. Why did you assume uniform distributions for the van Genuchten parameters or their logarithms?

We did that because we want to test over the entire physically plausible range of these parameters. We could also use a normal or lognormal distribution, but this would to a certain degree be like implying knowledge about the soil type. We will change the sentence to "The parameters included in the sensitivity analysis are listed in Table 3. They are sampled from uniform distribution functions ρ to cover their entire physically plausible range. Table 3 also shows the limits of these ranges. "

Table 4. To increase the readability of the Table independently from the text I suggest to clarify that the calculation times are for a PC in the first two cases and a supercomputer in the third case.

We will add this information.

Section 4.2 Could there be an effect of the way you calculate $r$ (Eq. (6)) on the accuracy of the non-iterative 2D model?

Yes, we agree with the reviewer that part of the mismatch with the non-iterative model could be an effect of our r-calculation method. As pointed out above, the method is rather ad-hoc and better ways may be possible, especially during the first coupling timesteps. Our used way of calculating r has though been showing the best overall performance in our test cases, which is why we use it. In the revised manuscript, the discussion about both the role of the non-iterative method as a comparison method rather than a suggested method, and our choice of r calculation method will be discussed in more detail.

l. 394. The spread in values of the specific yield is well beyond the plausible range. But since the specific yield is not really a parameter in your approach, I am not sure if that should be worrying or not.

In our opinion this is not a problem, which we try to explain in l. 469ff. Also, since this approach is novel there is no data, we could compare it to.

l. 407-407. I recommend that you mention the fact that the model is not designed to handle overland flow when you describe the models in section 2.

This is a good suggestion. We will add "The coupled model is a simplified subsurface model that can model flow in the groundwater and in the vadose zone, but no overland flow. It consists of a 2D depth averaged model for horizontal groundwater flow (Eq. 1) and multiple 1D models for vertical unsaturated flow (Eq. 2)." at the beginning of Section 2.2.1.

l. 461-465. You implemented unit gradient flow in the top of the profile and no flow (hydrostatic equilibrium) in the lower part of the profile, with an abrupt boundary between the two. In the first time step, this creates a rather hectic situation at the interface between these two regions, where Richards' equation needs to smoothen the transition and create some flow in the area with initially stagnant water. I can imagine this causes an initial jump in the importance of the van Genuchten parameters.

This is a good explanation. We will change l.464-465 to "This is most likely due to the rather artificial initial condition in the unsaturated zone which enforces unit gradient flow down to 1.25m above the groundwater table and no flow below."

l. 481-487. In your approach, the specific yield is no longer even a model parameter. The discussion of its physically impossible values of the specific yield is OK, but the rest of the paragraph is a bit contrived.

We partly agree and partly disagree with the reviewer here. The part was also commented on by reviewer 2. We will reformulate this paragraph and change it to:

"In the end, the specific yield is not a physical quantity here but a model parameter as it needs to compensate the influence of the lateral fluxes on the recharge, which cannot be quantified properly in this model, as well. Therefore, it should be treated as such and fitted for each application. This is a general problem, that does not only concern this model, as there is no good way of determining this parameter properly in advance. Calculating it within the iteration substitutes its calibration and is therefore an advantage over other methods. The sensitivity analysis for the parameters' influence on the calculated specific yield shows that it behaves reasonably. It depends on the parameters of the unsaturated zone models, especially on porosity $\phi$ and the saturated hydraulic conductivity $K_{UZ}$. The aim of the specific yield is to represent the missing unsaturated zone in the groundwater model, therefore a strong dependency on the unsaturated zone model's parameters is plausible."

l. 503-504. Unfortunately, the saturated hydraulic conductivity is also the parameter with the largest degree of spatial variation.

This is true but it needs to be estimated for the groundwater model anyhow.

---

## Author Response (AR1)

**Response to review #1**

Review of Brandthorst et al. 'Coupling saturated and unsaturated flow: comparing the iterative and the non-iterative approach'

We thank the reviewer for the effort and time to revise our manuscript and in the following want to respond in detail to the constructive comments he provided.

Major comments.

The contribution of the paper is purely computational. This can be seen from the test cases, all of which are highly artificial. I do not consider this a problem. The paper is focused, does not claim more than it delivers, and what it delivers is relevant for the HESS readership and substantial. The paper is well organized and generally clear. In the minor comments below I indicate where I was not sure if I understood everything.

We thank the reviewer for this rating. We agree that the test cases are entirely artificial. The aim was to apply and compare the methods under fully "controllable" conditions. Evaluating the performance of the models in a more realistic setting or even a real-world case would be interesting work for the future.

Minor comments.

The conclusion in the abstract that the more elaborate iterative coupling is better and more robust but slower than the non-iterative coupling is a bit underwhelming. Perhaps being more specific would make it more informative.

We reformulated this sentence to: "The non-iterative approach is faster but does not yield a good accuracy for the model parameters in all applied test cases whereas the iterative one gives good results in all cases. Which strategy is applied depends on the requirements: Computational speed vs. model accuracy."

The overview in the Introduction of the various coupled unsaturated-saturated models, their coupling strategies and the role of the specific yield is insightful and thorough.

Thanks.

l. 124- 131. If you assign the same 1D unsaturated zone columns to multiple grid cells of the underlying 2D groundwater model, I believe you are making the implicit assumption that the weather is uniform over the 2D extent of the groundwater level, or at least over the regions assigned to each of the 1D columns. If this is indeed the case, it may be good to mention this.

Yes, this is the case. We changed l.127-129 (now l.135-137) to "The atmospheric forcing of the unsaturated zone model as well as the recharge that is calculated from the unsaturated zone model and passed to the groundwater model are assumed to be constant within each zone. "

l. 156-173. If I understand correctly, if the resizing of an unsaturated zone column shortens the column, you have to add the storage in the part of the column that is cut off to the amount of water in the saturated zone to conserve mass during the resizing operation. When you increase the size of an unsaturated column, some water is transferred from the saturated to the unsaturated zone. I believe this is done in steps 4 and 5. In that case, I believe the ratio $r$ should normally be equal to 1 to ensure mass conservation. You give several reasons to deviate from this. Equation (6) gives the expression for $r$ you used, capped at 1. If you consider the entire system, does capping $r$ at a maximum of 1 whilst allowing values smaller than 1 not necessarily lead to mass losses when resizing all unsaturated columns?

Incidentally, for this reason I do not understand why Erdal et al. (2019) set r to 0 instead of 1, but I am not reviewing that paper here. But it seems to indicate that I missed something here, unless a shortening/lengthening of an unsaturated zone column led to a corresponding increase/decrease in its volumetric water content to keep the total amount of water present in the column unchanged.

Indeed, the r=0 assumption is questioned by us as well, and that is why the new ratio is used. One needs to consider that the models are not strongly coupled. Setting the ratio to 1 causes a lot of numerical problems following the one-step-behind adding of the lost water. An example is oscillating groundwater tables with increasing magnitude. We therefore settled for the rather ad hoc solution presented in the manuscript. As the capping occurs both on elongating and shrinking of the unsaturated zone, it will have both a positive and a negative effect on the total water in the system, so the comment of the reviewer is not quite right. We see that this was not well explained in the original submission and reformulated the corresponding paragraph (now l.170-185) to make this clearer. We also wrote explicitly that a factor r=1 would be needed to keep strict mass conservation.

Perhaps it is possible to have a 'before and after' figure, table, or water balance for the saturated and unsaturated domain (1 column only) in which you track what amounts of water go where during resizing.

While this is in principle an attractive idea, it should be noted that the non-iterative model, which, as the reviewer also points out, is taken from a previous publication on speeding up model spin-up (Erdal et al 2019), in this work serves mainly as a comparison to the iterative one. It is a conceptually quite ad hoc, but very fast model. Hence, we do not wish to spend more work on the inferior model, but rather focus on the much better performing iterative coupling, and we hope the reviewer can understand our point of view.

l. 158, 221. What is point i? Do you mean point 1?

Yes, it should be point 1. This is an artifact from the previous template using roman numerals. We corrected that.

Figure 2. Perhaps use a different color for the horizontal line denoting the phreatic level. (Minor point, please don't bother if this takes too much time.)

This is a good suggestion. We changed the color.

l. 203. Equation (10) forces the specific yield to conform to the simultaneous gain/loss of water and the resulting change in in the groundwater level. Essentially, the specific yield is no longer a model variable but the ratio of the calculated water gain per calculated groundwater level change. The water level change is calculated using the 1D Richards' equation, which does not have the specific yield as a parameter. You thereby to a certain extent impose the unsaturated zone model upon the groundwater model and adapt one groundwater model parameter (the specific yield) to match both models. Interesting approach.

Yes, this is exactly what we do and what to our knowledge has not been tested before. We think that the approach is well explained by the reviewer and partly adopted his formulation in the revised manuscript (now l.217-220).

l. 226. 'constant' or 'spatially uniform'?

We mean "spatially uniform". We changed this.

Figure 3. I was expecting a yes/no decision at the closure criterion, looping back to the start of the iteration in case of 'no'.

*Right, this was missing. We added it.*

l. 271-272. You state that you have hydrostatic equilibrium as the initial condition (zero flow), but in the next sentence state that you state that in most of the profile, the matric potential is vertically uniform, which amounts to a profile with unit gradient flow.

*We agree that the way it was written is misleading and changed the two sentences to: "The initial condition assumes a hydrostatic pressure profile in the lower 6.5m of the soil corresponding to an initial position of the groundwater table at 3.95m below the surface. In the upper 3.5m of the soil unit gradient flow is imposed by setting the pressure head to -0.283m. "*

l. 281-282: repeats l. 276-277.

*Not exactly. L.281 – 282 refer to the 1D reference model and l .276-277 to the coupled model. We made that clearer in the revised manuscript (now l.317-322).*

l. 325. Why did you assume uniform distributions for the van Genuchten parameters or their logarithms?

*We did that because we want to test over the entire physically plausible range of these parameters. We could also use a normal or lognormal distribution, but this would to a certain degree be like implying knowledge about the soil type. We changed the sentence to "The parameters included in the sensitivity analysis are listed in Table 3. (…) The parameters are sampled from uniform distribution functions $\rho$ to cover their entire physically plausible range. Table 3 shows the limits of these ranges. "*

Table 4. To increase the readability of the Table independently from the text I suggest to clarify that the calculation times are for a PC in the first two cases and a supercomputer in the third case.

*We added this information.*

Section 4.2 Could there be an effect of the way you calculate $r$ (Eq. (6)) on the accuracy of the non-iterative 2D model?

*Yes, we agree with the reviewer that part of the mismatch with the non-iterative model could be an effect of our r-calculation method. As pointed out above, the method is rather ad-hoc and better ways may be possible, especially during the first coupling timesteps. Our used way of calculating r has though been showing the best overall performance in our test cases, which is why we use it. In the revised manuscript, the discussion about both the role of the non-iterative method as a comparison method rather than a suggested method (see Section 4.5 and 5), and our choice of r calculation method (now l.170-185) are discussed in more detail.*

l. 394. The spread in values of the specific yield is well beyond the plausible range. But since the specific yield is not really a parameter in your approach, I am not sure if that should be worrying or not.

*In our opinion this is not a problem. However, after conducting a convergence study due to the comments by reviewer 2 we refined the grid for this test case. This reduced the spread of specific yield values significantly and we do not get values out of their plausible range anymore.*

l. 407-407. I recommend that you mention the fact that the model is not designed to handle overland flow when you describe the models in section 2.

This is a good suggestion. We added "The coupled model is a simplified subsurface model that can model flow in the groundwater and in the vadose zone, but no overland flow. It consists of a 2D depth averaged model for horizontal groundwater flow (Eq. 1) and multiple 1D models for vertical unsaturated flow (Eq. 2)." at the beginning of Section 2.2.1.

l. 461-465. You implemented unit gradient flow in the top of the profile and no flow (hydrostatic equilibrium) in the lower part of the profile, with an abrupt boundary between the two. In the first time step, this creates a rather hectic situation at the interface between these two regions, where Richards' equation needs to smoothen the transition and create some flow in the area with initially stagnant water. I can imagine this causes an initial jump in the importance of the van Genuchten parameters.

This is a good explanation. We changed l.464-465 (now l.533-534) to "This is most likely due to the rather artificial initial condition in the unsaturated zone which enforces unit gradient flow down to 1.25m above the groundwater table and no flow below."

l. 481-487. In your approach, the specific yield is no longer even a model parameter. The discussion of its physically impossible values of the specific yield is OK, but the rest of the paragraph is a bit contrived.

This part was also commented on by reviewer 2. As we do not get these unphysical values for the specific yield anymore, our discussion here became obsolete. We shortened and reformulated the entire paragraph (now l.537-546).

l. 503-504. Unfortunately, the saturated hydraulic conductivity is also the parameter with the largest degree of spatial variation.

This is true but it needs to be estimated for the groundwater model anyhow.

**Response to review #2**

Referee comment on "Coupling saturated and unsaturated flow: comparing the iterative and the non-iterative approach" by Natascha Brandhorst et al., Hydrol. Earth Syst. Sci. Discuss., https://doi.org/10.5194/hess-2021-15-RC2, 2021

The manuscript presents two methodologies of coupling for dimensionally heterogeneous modelling of subsurface flow in the unsaturated zone and in the saturated zone. The interest of such an approach is that important diminutions of computation times may be obtained compared to fully 3D modeling approaches. The main drawback is that accuracies of the simulations are dommaged, still in comparison with 3D modelling, and more or less along the considered cases and coupling methodologies. Given the very large computation times that may be encountered in fully mechanistic hydrological modeling at the wtareshed scale, this problem is of great interest for the community of hydrological modeling. The manuscript contains an important material in terms of numerical results and provides relevant hints to compare the two coupling strategies under concern.
Nevertheless the presentation of the considered theories and numerical experiments lack of rigor, and the writting of the manuscript is not clear enough. In some places additional computations may even be needed. Thus I think it should be thoroughfully reworks prior to publication. I recommend to reject the paper in its present form, and to encourage the authors to resubmit after having completing and improving it.

We thank the reviewer for the effort and time to revise our manuscript. He provided many helpful comments and remarks that need to be reworked by us. However, we do not understand his suggestion to reject the manuscript based on his comments which mainly require further explanations and only a few additional computations. In the following, we will answer to all comments in detail and hope that we can thus clarify all open points, but also underline our point of view where we do not agree with the reviewer.

**General comments:**

- The considered equations should be defined more rigourously and rewritten. For instance, the double time derivative term in equation (2) is a non-standard formulation of Richards equation (see for instance Gottardi and Venutelli, 1993). I guess that considerations related to the order on magnitudes of those two time derivatives may be used to justify the adopted formulation, but it should be explicited. Moreover, the use of the same notation $S_y$ for the specific yield in equation (1), which has classically a clear and well identified physical meaning (drainage porosity), and for the iteratively computed, time variable fitting parameter used in the iterative method to handle recharge fluxes from the unsaturated zone is confusing. I think that the latter one should be expressed as the sum of the true specific yield $S_y$ and a new additional term used for the purpose of the coupling between the saturated zone and the non-saturated zone. This would not imply new computation, but simply to rewrite some equations and rescale some results. I think that the added value of such more accurate notations in terms of clarity and of ease of physical interpretation would be important.

We will answer this comment separately for equation (1) and (2). The comment implies that there is a general issue concerning more equations, but as no other equations are mentioned explicitly and we do not see any problems with our formulations we will limit our reply to these two:

Equation (1): This formulation was also noted by the third reviewer, so we will give the same explanation here:

The left member of Eq. 2 is derived from the time derivative in the volume balance (we assume incompressibility of water):

$$\frac{\partial V_w}{\partial t} = \frac{\partial (S(h_p)\phi V_t)}{\partial t} = \phi V_t \frac{\partial S(h_p)}{\partial t} + S(h_p)V_t \frac{\partial \phi}{\partial t} + S(h_p)\phi \frac{\partial V_t}{\partial t}$$

$$= V_t \cdot \left( \phi \frac{\partial S(h_p)}{\partial t} + S(h_p)\frac{\partial \phi}{\partial t} + \frac{1}{V_t} S(h_p)\phi \frac{\partial V_t}{\partial t} \right)$$

$$= V_t \cdot \left( \frac{\partial (S(h_p)\phi)}{\partial t} + S(h_p)\frac{\phi}{V_t}\frac{\partial V_t}{\partial h_p}\frac{\partial h_p}{\partial t} \right)$$

$$= V_t \cdot \left( \frac{\partial (S(h_p)\phi)}{\partial t} + S(h_p)S_s \frac{\partial h_p}{\partial t} \right)$$

with subscripts $w$ and $t$ denoting *water* and *total.*

The specific storage is in general defined as $S_s = \frac{1}{V_t}\frac{\partial V_p}{\partial h_p} = \frac{\partial \phi}{\partial h_p} + \frac{\phi}{V_t}\frac{\partial V_t}{\partial h_p}$ with $V_p = \phi V_t$ being the pore volume. This definition is used when the change of porosity is written in dependence of the change in water pressure head (so $\frac{\partial \phi}{\partial t} = \frac{\partial \phi}{\partial h_p}\frac{\partial h_p}{\partial t}$). One then also gets a slightly different formulation of Eq. 2: $\frac{\partial V_w}{\partial t} = V_t \cdot \left( \phi \frac{\partial S(h_p)}{\partial t} + S(h_p)S_s \frac{\partial h_p}{\partial t} \right)$.

In our case, we maintain the time derivative of porosity (or rather water content, as $S(h_p)\phi = \theta$) and therefore the specific storage formulation reduces to $S_s = \frac{\phi}{V_t}\frac{\partial V_t}{\partial h_p}$, which is equivalent to assuming that $\frac{\partial \phi}{\partial h_p} \cong 0$.

This formulation of Richards' equation can be found in e.g., Kavetski et al. (2001), Kollet and Maxwell (2006); Fahs et al. (2009). We understand the reviewer's comment that there are more common formulations of this equation and therefore, in the revised manuscript, clarified the definition of the specific storage, motivated our choice to use this formulation and gave a reference (now l.113-120).

Equation (2): Here, we disagree, but we understand the comment since the specific yield in equation (2) is often set equal to drainage porosity, which we mention in the revised manuscript (now l.65-66). The problem is that this definition neglects the soil water stored above the groundwater table and thus overestimates the free capacity for groundwater storage. However, using drainage porosity is kind of an "established" way since it can be accurate enough in some applications and there is simply no better way of estimating this parameter without modelling or, in case of field studies, monitoring the water content in the unsaturated zone. When deriving the 2D groundwater flow equation for an unconfined aquifer from the volume balance (we assume incompressibility of water), the specific yield turns out to be equal to

$$S_y = \frac{\partial V(h)}{\partial h} = \frac{\partial}{\partial h}\left( \int_{z\_bottom}^{z\_surface} \theta(h,z)dz \right)$$

to ensure mass conservation. h is the water pressure head. This can be close to drainage porosity under specific conditions, but in general it is not and can be even far off, depending on the soil moisture profile. This dependency is no new finding and has been described by others (see the related literature cited in the introductory part of our manuscript (now l.63-87)) who also object to defining the specific yield simply as drainage porosity. We see that depending on the application this definition might be sufficient and not cause any larger problems. However, in our case (where we have information about the soil moisture profile) a definition as drainage porosity would mean violating the conservation of mass. By the adaptive determination of this value during the iterations, we use this information to get a better estimate of this "true" specific yield. Unfortunately, we cannot get there entirely due to the influence of the lateral fluxes on the recharge as we explain in l. 222-228 (now l.236-240). It is important to us that it is understood that this is not a mere coupling

parameter for handling the recharge fluxes in our model but is as close as we can get to the value of this.

Concerning the notation, we could indeed write the specific yield as the sum of two terms, the drainage porosity and the part that takes the soil moisture into account. However, we preferred not to do this in order to be consistent with other papers that are relevant for our work (e.g., Pikul et al., 1974; Crosbie et al., 2005). In these papers, $S_y$ as specific yield is also not used as drainage porosity.

- Convergence studies for mesh refinement and time stepping strategy are not evocated as it should be the case in any study producing Computational Fluid Dynamics results. In some places it may imper the possibility to understand the comparative behaviours of the proposed test cases. For instance if we consider the comparison of accuracies of test 2 and test 3, in the present form of the manuscript it is impossible to say what comes from the differences of meshes and what comes from the different physics under concern (e.g.: homogeneous versus heterogeneous soil).

This is a fair comment and we agree that grid convergence should be addressed. We performed convergence studies testing the general convergence behavior of the models before and now did a convergence study with the iteratively coupled model for the three applied test cases. Since the first test case is 1D, convergence of the horizontal grid was only tested for the other two cases. Figure 1 shows the results. Beegum et al. (2018) use a grid resolution of $\Delta x=100m$ (red circle) which corresponds to a mean absolute error (of groundwater tables over time) of MAE$\approx5\cdot10^{-3}$m. As we want to compare our results to theirs, we chose our grid sizes accordingly. Thus, for the third test case we had to use an even finer grid ($\Delta x=5m$) than in the original version.

[Figure]

*Figure 1: Grid convergence for Δx=Δy for the second (left) and third (right) test case.*

[Figure]

[Figure]

[Figure]

*Figure 2: Grid convergence for Δz for the first (upper left), the second (upper right) and third (lower) test case.*

*The same applies for the vertical direction. Here, 200 instead of originally 50 nodes are required to achieve the chosen accuracy (see Figure 2). As the grid spacing in this test case is non-uniform, we did the convergence study based on the number of nodes instead of the cell size Δz for this case. In the second test case, on the other side, we saw that a coarser vertical grid with Δz=0.2m instead of Δz=0.1m is sufficient. For the first test case, we could keep the original resolution.*

We also performed a convergence study on the coupling time step size but did not see any influence. Therefore, we kept the original time stepping setup.

We added a short paragraph concerning grid convergence in Section 3 (now l.293-307) as we do not want to make the manuscript too long. To keep the comparability among the coupling approaches and the reference model we used the same grid for all compared models. One must keep in mind, though, that the vertical grid in the non-iteratively coupled model changes over time, so we can only set the initial grid accordingly.
After the convergence study, we can now be sure that the different accuracies for test case 2 and 3, are not due to the finer mesh used in the third test case.

- To the knowledge of the reviewer, an important example of hydrological model that couples dimensionnaly heterogeneous descriptions of flow in the saturated zone and in the unsaturated zone is MIKE-SHE (e.g.: Graham and Butts, 2005), which is for instance included in recent international benchmarking efforts for physically based hydrological modeling (e.g.: Kollet *et al.*, 2016). The fact that works related to MIKE-SHE do not appear in the references of the manuscript make me think that the bibliographical survey on which the presentation of the background of the study is done should be consolidated.

We are fully aware of the MIKE-SHE model and included it in our literature survey (now l.23-27, 29-32 and 51-53). Its coupling strategy does not differ substantially from the models we already referred to (no overlap of the two compartments, compartments solved separately, iterative procedure with step-wise adjustment of water table to improve mass balance), so we added the reference as another example.

**Specific comments:**

- l 136-137: "a Neumann boundary representing net flux from precipitation and evapotranspiration" : with the source/sink term of the equation (2), it is possible to represent actual evapotranspiration distributed in time and space according to water avalaibility in the soil (see for instance Orgogozo et al., 2019) ; please discuss the limitation associated with an a priori estimation of the actual evapotranspiration directly embedded in the Neumann boundary.

We agree that the source/sink term can be used to represent actual evapotranspiration in the root zone. Here, we do it in a simplified way merging the net fluxes into the Neumann boundary at the top because we have no information on the root zone distribution or vegetation. We could assume one, but the good match of the simplified approach and the results presented in Beegum et al. (2018) show that it is not needed. In the other two test cases there is only precipitation and evaporation does not play a role. Also, our focus is on the groundwater table. The implementation of evaporation would be more important if the focus would be on land surface processes. We changed the sentence to "The upper boundary condition of the unsaturated zone models is a Neumann boundary representing the flux across the land surface. In this work, the flux will be either precipitation or the net flux from precipitation and potential evapotranspiration depending on the test case." to make the boundary condition clearer.

- l 151 : "collect the computed recharge (i.e. flux leaving over the bottom boundary) and interpolate the 2D map of groundwater recharge." : You mean collect all the computed regarges for all time steps of the 1D Richards model since the previous time step of coupling ? Should be clarified.

*Indeed, that is what we mean. This was made more explicit in the revised manuscript (now l.159-161).*

- l 157 : "Add (or subtract) a ratio r of this water to the recharge computed in the next time step." you mean the next time step of coupling ? Should be explicited.

*Indeed, that is what we mean. We changed this sentence to "Add (or subtract) a ratio r of this water to the recharge computed in the next coupling time step." to make it more explicit.*

- l 159 – 171 : The proposed way of chosing the ratio r is difficult to accept. In case of water table elevation, the ratio r could be fitted to keep unchanged across the mesh resizing process the total amount of water contained in the part of the domain that stays unsaturated, while in case of water table lowering a 'field capacity' water saturation could be prescribed to the cells that experienced desaturation in order to compute a total water amount to be distributed in the new 1D mesh, with an associated proper r ratio? Here the formulae proposed for the computation of the ratio r seems somewhat arbitrary. For instance the point (1) "the groundwater table rise or fall is also effected by lateral flows" is already taken into account in the 2D groundwater model. Besides, "the unsaturated zone is really compacted by a rise of groundwater levels" does not sound physical at all.

*In principle we do not disagree with the reviewer on this point. The non-iterative method consists to a fair degree of ad-hoc solutions that originate from the aim to keep the model as simple as possible while maintaining acceptable results. In this, we found the suggested r-ratio to give a reasonable result. A lot of testing, similar to those suggested by the reviewer have been done on the model, but the sparse representation of the unsaturated zone often causes practical problems with refined methods. Nonetheless, it should clearly be noted here that the non-iterative model, which is taken from a previous publication on speeding up model spin-up (Erdal et al 2019), in this work serves mainly as a comparison to the iterative one. This is the new method for this publication and the better one, and the one we recommend using. We made that clearer in the revised manuscript (see Section 4.5 and 5) and addressed the reviewer's remarks on the formulations (now l.170-185).*

- l 183 : "(v) the iteration counter" : with which loop is related this iteration counter is unclear at this point (it could be for instance with the time stepping of the 1D Richards equation or with the coupling time steping)? Although it becomes clear afterward, it should be explicited here, at the first occurence of (v).

*We agree that this can be misleading at this point. We added the information by changing the sentence to "… and the superscript (v) the iteration counter for the coupling time step loop."*

- l 203 eq (10) (see the first general comment): According to the basic derivation of the diffusivity equation for unconfined aquifer, the specific yield is equal to the drainage porosity of the considered porous medium – although it seems that it might be different for more elaborated derivations, according to the literature cited by the authors. What is the physical interpretation of the variations the specific yield computed by eq (10)? Is there a theoritical reason why the iterating on the values of the specific yield field in the aquifer should lead to convergence? In case this is a purely empirical methodology, are there cases for which divergence may occur? Other questions : the value of the 'physical' $S_y$ parameters that appears in the equation (1) is only the seed of the iterative process at the first time step of simulation, and do not appear directly anymore in the course of the simulation for the evaluation of $S_y$ $_v$, right? That is what I understand from table 2 for instance. It should be clarified here.

*The first part of this comment is already answered in the general part so we will focus here on the remaining questions.*

(a) Regarding the convergence of the specific yield: There can only be a consistent solution for the coupled model if the specific yield converges. As in any iterative process the success of this iteration depends on the chosen method, starting value and closure criterion. Divergence can occur if the model error is too large. This could mean here: too coarse grid, too large (coupling) time steps, too few unsaturated zone models, large differences in the hydraulic conductivity in the saturated and unsaturated model (due to the averaging). We added a sentence in Section 2.2.3 (now l.221-223) where we discuss this.

(b) Regarding the value of $S_y$ given in table 2: This is indeed only the seed. This is stated in the footnote of table 2 and shown in Figure 3.

- l 221 – 224 : "The source/sink terms $q_{lat,i}$ have an effect on the recharge ('R(Q lat = 0) ≠ R(Q lat ≠ 0)'), which due to the nonlinearity of Richards equation cannot be quantified." However at step (4) (l 216 – 217), an updated $R_v$ is computed that takes into account the $q_{lat,I\ v}$ ? I don't understand.

We agree that this is confusing. We cannot quantify the effect of the lateral fluxes on the recharge, so we keep the water table fluctuation caused by the fluxes in the unsaturated zone during one coupling time step $\Delta H_{UZ}{}^1$ constant. The recharge R is only changed because the specific yield is updated during the iterations (see equation (7)). We added the sentence "Changes of the recharge $R^v$ during the iteration are then only caused by changes of the specific yield (see Eq. 7)." (now l.239-240).

- l 229 – part 2.3 Activity score: Difficult to follow. Lack of explanations and of references. There is also a problem of structure: since 'The parameters and the model output f are defined in Sections 3.4 and 4.4, respectively.', this part 2.3 is not possible to understand by itself at this point of the reading. This part should be reworked so that the reader may understand why it is interesting to use the activity scores for the sensitivity study, and on what motivated the choices of the parameters x and the output f(x). By the way, in table 3 part 3.4, the variable $K_{GW}$ and $K_{UZ}$ seems not to be defined in the manuscript ? And why chosing $S_y$ as a parameter of the sensitivity study while it is subjected to iterative evolution of its value along computation in the iterative method (l 218 – table 2)?

In a previous version of the manuscript this part was more detailed, but we considered it too long and taking too much space, so we shortened. We see that we overdid the shortening and put more explanation back in (now l.245-276). We also added a list of the used parameters and model output already in this section to make it easier to understand. The parameters $K_{GW}$ and $K_{UZ}$ are now introduced properly at their first appearance in Section 3.4.
The aim of the sensitivity analysis is to reveal the most influential parameters. As we claim that the specific yield has a large impact on the model output, we need to include it in the sensitivity analysis to confirm this statement. The goal of the sensitivity analysis is also to check if the not fully coupled models do not generate unphysical behavior in the sense that parameters influence the properties they should influence. We made this clearer in the revised manuscript (now l.375-377).

- l 267 : '3.1 Test case 1: 1D flow' lack of a figure that presents the geometry, the boundary conditions and the meshes for each models.

Agreed. We provided such a figure.

- Figure 4 : Wrong title for y-axis (Precipitation-PET, not just PET)

The title is correct. We are aware that the abbreviation PET is reserved for potential evapotranspiration in some communities, but we chose to define it otherwise (precipitation evapotranspiration, see l.256). To avoid confusion, we changed the term to P-ET (now l.310).

- l 276 : Precise which 1D model (pure Richards I suppose ? )

Yes, this is a pure Richard's model. We stated this is in l. 261-262 (now l.287-288), but we mention it here again in the revised manuscript (now l.317) to avoid any misunderstanding.

- l 281 : "The groundwater domain is divided into a 2 × 2 grid. Each groundwater cell is assigned a 1D model." Then the groundwater model is 2D with only 2 cells in each direction ? I don't understand.

Yes, the groundwater model is 2D (we used the same grid as in Beegum et al. (2018) for comparison) but due to the soil structure and boundary conditions, flow is only vertical. This is similar to test case 2, which is 3D but only has 2D flow.

- l 285 : "Since there is no variability along the 8000 m side, flow is effectively 2D in this test case." Then it is useless and misleading to present it as a 3D computation ; the figures and the discussions should be reshape for presenting directly the test case as a 2D one. The comparison of computation times is also questionnable : to deal with a 2D case in 3D increase tremendously (and artificially) the computation time with a fully mechanistic 3D model. Here some additionnal simulations (dealing with the 2D problem in 2D) are needed for making the comparison of computation times.

We disagree to this comment. Firstly, we do not claim that this is a 3D computation, the title of this subsection says clearly that it is a 2D problem. Secondly, we solve it in 3D, so giving 2D figures would be wrong. We agree that there is no real point in using a 3D model for this, but again (as in the previous comment) we would like to state that we did not design this test case (this was done by Morway et al., 2013 and Beegum et al., 2018) but only apply it to compare with already published results. Besides, the comparison of computation times is still valid as we use the same 3D setup for all models (the two coupled ones and Parflow). We want to stress here too that we do not intend to determine exact values for the computation time, which will anyhow strongly depend on the computational resources, but are only interested in the comparison between the models.

- l 290 : "[...] assigning a minimal initial pressure head of −1.25 m" ; you mean that -1.25 m is the pressure head at the top of the domain ? Please clarify.

We mean that the pressure head cannot have a value lower than -1.25m. Our formulation was not clear here. We changed it to "applying hydrostatic equilibrium and assigning a minimal initial pressure head of -1.25m at locations where the pressure head is below this value."

l 291 : "Monthly varying rainfall (Fig. 6) is used as Neumann boundary condition for the land surface". More precision about these data would be useful – e.g.: are they synthetic ? Of which type of climate are they representative ?

The data is taken from Morway et al. (2013) and information about the origin of the data is not given in their publication. We assume that it is purely synthetic but cannot be sure and therefore do not want to give any information here that may be wrong in the end.

- l 292 : Table 2 is not timely introduced ; since it contains information for the 3 test cases, it should be placed either in the beginning or at the end of the presentation of the considered test cases, but not at the middle.

We agree, and changed its position to the beginning of this part.

- l 293 – 294 : "grid size Δx = Δy = 100 m and Δz =0.1 m.". It makes a form factor of $10^3$ ... Any convergence study done for the mesh refinement?

In our opinion the form factor is not so unrealistic, as the 1D models need a much finer grid for convergence. The grid chosen based on the convergence study has a form factor of 500.

- l 296 : "With the flow problem being 2D this means that the entire domain is actually covered by 1D models." Nevertheless as far as I understood the proposed methodologies it would be exactly the same if the case was a 3D one? And I don't understand to which extent a 1D approximation for a 2D problem would be essentially more "acute" than a 1D approximation for a 3D problem?

For a 3D case, we would need 80x40 (160x80 with the updated grid) 1D models to cover the entire domain, as conditions would vary along the 8000m side. The word "acute" was a typo and should be "actually", we corrected that.

- l 306-307 : "three different soil units are distributed throughout the domain as depicted in Fig. 7." More information is needed here. Is this distribution synthetic? How has it been acquired / built ? Of which type of soil (sand, loam, clay ...) each unit is representative ?

The distribution is synthetic. The units 1 to 3 are representative for loam, loamy sand and sandy clay. We added this information in Table 2, Figure 8, and l.346-348.

- l 308 : "averaged arithmetically" Any tests for the use of harmonic or geometric mean instead of arithmetic mean?

No, from our point of view such tests are not needed. As we only have lateral flow in the groundwater domain, we can consider this problem as a horizontal flow through a vertically layered (stratified) medium. For such a setting, the arithmetic average is the adequate choice for all effective parameters.

- l 309 : " In the vertical direction a non-uniform grid is used with smaller grid sizes close to the surface and a total of 50 cells." Please provide the minimum and maximum sizes.

The grid size in the vertical direction ranges between and 0.75mm and 0.3m. We added this information (now l.352-353).

- l 315 : "The 1D models are placed at the center of each zone." How are laterally averaged the porous medium properties in each 1D models covering 10*8 cells laterally?

They are not averaged but taken at the specific location. We did not mention this and added this information (now l.356-357).

- l 327 : "The residual saturation $S_r = \theta_r/\theta_s$ and the specific storage $S_s$ are excluded from the analysis and set to 0.01 and 0.0015, respectively." Why have they been excluded ? To be justified, or at least discussed.

We complemented this sentence by "…, as a previous smaller sensitivity analysis had shown no impact of these parameters (not shown)."

- Table 3 : The parameters $K_{GW}$ and $K_{UZ}$ are appearing in the manuscript for the first time in this table. The notations used in table 3 and those used in the equations (especially (1) and (2)) should be the same, of at least explicitly related.

We agree. As we already mentioned in our reply to a previous comment, we introduced these parameters properly at their first appearance in the revised manuscript.

- l 329 -330 : "The horizontal spatial resolution is again $\Delta x = \Delta y = 10$ m, whereas the vertical resolution is $\Delta z = 0.1$ m as in the 2D flow case." Once again a convergence study must have been done to justify the use of this mesh with a form factor of $10_2$.

We do not see a problem with the form factor, as the 2D (x-y) groundwater flow and the 1D (z) vadose zone flow are decoupled. We did not perform a convergence study for this test case but chose a grid resolution that is between that of the second and third test case since this setup is a mixture of the other two. The convergence study for the second and third test case have shown that such large form factors are required to achieve the target accuracy.

- l 332 : "The time step sizes are the same as in the previously described test cases." Any convergence study for justifying the use of the proposed time stepping policy?

The convergence study on the time step size revealed a negligible influence of the time step size (see reply to a previous comment about grid convergence).

- l 345 : "A visual comparison indicates that the coupling applied by Beegum et al. (2018) yields a comparable accuracy." Why not plotting the results of Beegum in Figure 8?

We agree, but we did not plot the data because we do not have them. We could put their figure next to ours if this helps, but we would have to clarify copyright issues for this.

- l 356 Table 4: This table contains information for all test cases and then it is not at the right place, being presented in a part specific to test case 1. Besides, since in test case 1 there is no lateral flux and thus no iteration in the iterative methods, I wonder why the iterative method has a wall time twice time more long than non-iterative method, while this later one include an addtionel step of remeshing? To be discussed.

We moved the table to the beginning of this section. The reason why the iterative model needs more time is because the grid is larger (including the saturated zone). We added the sentence "This is due to the larger grid of the iterative model which also includes the saturated part." at the end of this paragraph for explanation.

- l 363-364 : "The results by Beegum et al. (2018) have a similar accuracy and shape as the results of the iterative coupling approach." Why not plotting them in Figure 9?

See reply to the previous comment: Because we do not have the data. We could put their figure next to ours if this helps, but first would need to clarify the copyright issues.

- l 365 : "When considering the non-iterative model, it is notable that initial time steps are an issue [...]" Any convergence study on time step ? What happens if smaller time steps are used?

We have extensively tested the case with smaller time steps, but the general behavior still remains, and does not seem to be an issue with time step size. This information was clearly missing in the original submission and was added in the revised manuscript (now l.415-416).

l 366-368 : "Both of these issues may be related to the reference model essentially acting as a bucket without any plausible steady state solution (i.e. steady state for the groundwater model would have groundwater tables far above the top of the domain)." Then why not chosing lower values of precipitation , so that a steady state may be reached?

We use this test case to compare to the results of Beegum et al. (2018), so it would not make sense to change the precipitation.

- l 374 : "All values [of $S_{y\,0}$] are smaller than the proposed value of 0.28, although the difference is less than 0.03." How the proposed value of 0.28 has been choosen? Are there correlations between the $S_y$ values and the state of the groundwaters (e.g.: water table altitude, lateral fluxes intensity)?

The value of 0.28 was taken from Beegum et al. (2018), who do not justify their choice. We assume that they fitted it beforehand. And yes, there are correlations. There is a linear relationship between the water table position and the Sy values in this case (so Sy values increase with increasing water table position). We added a figure (Fig.11, right plot) showing this relation, but we also want to point out that this relationship is probably case specific and should not be overrated. For the third test case we saw a different relation/dependence. A figure (Fig.15, right plot) and a short discussion of that (now l.452-458) are added in the revised manuscript, too.

- l 379 : "Both coupling schemes show a good agreement with the fully integrated 3D model." It is hard to understand why the matching between the fully 3D computation and the 2.5D ones is better here for this 3D heterogeneous test case than in the 2D homogeneous test case 2. I noted that in test case 3 a finer mesh is used than in test case 2. May be that convergence issues are at stake?

This is explained in l. 452-459 (now l.520-527). It is caused by the boundary condition. We refer to that section here in the revised manuscript (now l.437-438), so that the reader knows an explanation will come later in the discussion part. After the convergence study we can exclude possible effects of the grid resolution.

- l 389-391 : "Areas with larger differences appear at similar locations for both coupling schemes showing the largest deviations of up to ΔH GW = 1.5 m along the y = 800 m boundary". Why are they such discrepancies, and why there? These points should be discussed here.

We thank the reviewer for pointing us to this part. Doing the analysis, we noticed that we made a mistake during the postprocessing of the data. The larger discrepancies occur close to the lower Dirichlet boundary (so between x=300m and x=400m). These differences are due to the zonation and the resulting simplified recharge pattern which hast a larger impact in this part of the domain. We corrected the figure and added another one (Fig.14) for explanation. We also adjusted the corresponding discussion part (now l.439-452 and 527-528).

- l 394-396 : "Overall, the specific yield values are decreasing when the groundwater table is rising and increasing when the groundwater table is falling (roughly between t = 1100 d and t = 2200 d)." Once more, a careful discussion of the physical meaning of $S_y$ and its variation is needed.

The question was raised also by reviewer 3 and we copy here the answer to his / her comment:
One cannot really say at this point what is expected because the heterogeneities have a strong impact which is hard to evaluate. Further analysis during the revision showed that there is a strong dependence on the soil properties at the water table position. We show this in the revised manuscript (now l.452-458). A more detailed discussion on the Sy values is not possible without more tests.

- l 414 : "Note that the specific yield in the iteratively coupled model is not the value used for the non-iteratively coupled model defined in Table 3 but the value calculated by the model during the simulation." I don't understand how it is possible to make a sensitivity analysis on a parameter that is not constant and specified prior to computation, but timevariable, calculated along computation?

The activity scores are calculated individually for each time step, therefore a time variable parameter as input is no problem. We hope that this is clearer in the revised manuscript where we explain the method of activity scores better in the second section.

- l 417 : "Acitivity"

Yes, this was a typo, and we corrected it.

- l 421-422 : "When looking at Eqs. 1 and 7, one sees that $S_y$ can be eliminated which explains why there is no influence of $S_y$ under these conditions." You mean that dh/dt = 0 at extremas ? To be clarified.

We changed the sentence to "When looking at Eqs. 1 and 7, one sees that $S_y$ can be eliminated when lateral fluxes are negligible compared to the recharge fluxes, which explains why there is no influence of $S_y$ under these conditions." for clarification.

- l 431-432 : "The average $S_y$ value shows some smaller fluctuations, but overall it converges to a value around $S_y$ = 0.17, which is a plausible value.". This is a too short discussion of the value of this key parameter that controls the exchanges between the saturated zone and the unsaturated zone in the iterative method. It should be interpreted physically. It seems to potentially encompasses a non clearly identified list of physical phenomena.

We do not really understand the need for a discussion at this point. It is already interpreted physically and a further discussion of the value is not possible.

- l 436-437 : "This means that the specific yield is mainly depending on the unsaturated zone parameters. This is reasonable as its intention is to represent the missing unsaturated zone in the groundwater model." Somewhat strange. According to the basic derivation of the diffusivity equation for unconfined aquifers, the specific yield should be a property of the saturated zone (drainage porosity). So may be that if this parameters depends mainly on the properties of the unsaturated zone, it means that it is not, or not only, a specific yield (see the first general comment)?

As outlined in our reply to the general comment, we disagree to this interpretation of the specific yield. We would argue that the specific yield is not a property of the saturated zone but a property of the unsaturated zone which is used in the saturated zone model. We refer to the reply to the general comment.

- l 442 : "in the case of the iterative model even consistent." I am not sure of what you want to say, please be more specific.

We reformulated to "… and the model compartments are consistent."

- l 448 : "On the contrary, using more models could help decreasing the discrepancies in the less accurate areas close to the no-flow boundary at y = 800 m which are most likely caused by the soil heterogeneities and the simplified recharge and specific yield pattern due to the zonation." These discrepancies are important (~1,5m), and their causes must be carefully assessed. Additional numerical experiment with lower and stronger soil heterogeneities or various zonation startegies could help to ensure that the proposed diagnostic is correct. From my point of view stating that "As this is a general issue for these kind of models and does not relate to the presented coupling strategies themselves, we do not investigate it further." is not sufficient, at least without any bibliographical references as it is at present.

Since we now found that the discrepancies do not occur at that location, but at the lower Dirichlet boundary (which is explained at the comment on l. 389-391), this discussion part is not needed any more and we simply removed it. Otherwise, we would have agreed that further investigation would be necessary here.

- l 458-459 : "Therefore the results of the coupled model are on average more accurate even though this test case is more complex than the 2D flow case." Meshes also are different, and without proper convergence studies the impact of this point may not be

assessed. The convergence studies must be done, and used for consolidating the discussions.

We did the convergence study and found that these differences are not due to differences in the mesh.

- l 464 : "is constantly $h_{UZ} = -1.25$ m at $\geq 1.25$ m above the groundwater table" This should be made clear sooner (see teh comment on l 290).

Yes, we agree and refer to our reply to the comment on l 290, too.

- l 467 : "Which parameter is dominating depends on the current flow conditions." This should be discussed in more details.

This is discussed in detail in l. 417-427 (now l.478-488). We changed this sentence to "Which parameter is dominating depends on the current flow conditions as described in Section 4.4."

- l 470 : "comformably" is not specific/quantitative enough.

We changed to "consistently".

- l 478-480 : "This is not the case in this model as we cannot calculate this effect properly and we therefore keep ΔH due to recharge fixed (see Eq. 7)." However in equations (7), (8) and (10), it is clear that there is an iterative procedure that involves $\Delta H_{UZ}$ $_v$ and $\Delta H_{GW}$ $_v$ that evolve at each iteration $v$? I don't understand.

This comment is answered by our reply to the comment on l. 221-224. $\Delta H_{UZ}$ and $\Delta H_{GW}$ are changed during the iteration, but $\Delta H_{UZ}$ consists basically of two components: A change caused by lateral fluxes and a change caused by the recharge. The latter is what we refer to here and this is fixed. The fluctuations due to lateral fluxes change during the iteration and thus also the total water table fluctuation. We tried to explain this better in the revised manuscript (now l.236-240 and 541-542).

- l 481 : "In the end, the specific yield is not a physical quantity but a model parameter.". This statement seems too general ; while it is clearly the case in the proposed modeling approach, it is not the case in all formulation of the diffusivity equation in unconfined aquifers. Overal all this paragraph should be rewritten to better discuss the meaning of the concepts that are specific to the proposed methodology with a wording that should not rise ambiguities between these concepts and previously existing concepts. For instance:

This part was also commented on by reviewer 1. We shortened and reformulated the entire paragraph (now l.537-546).

- l 485-486 : "The aim of the specific yield is to represent the missing unsaturated zone in the groundwater model" You are talking about what you called a specific yield in your model. I think that it should have another name that 'specific yield', this latter word designing a concept that do have physical meaning and that is related to the properties (drainage porosity) within the saturated zone in the basic form of the diffusivity equation for unconfined aquifers (see the first general comment).

We disagree to this interpretation and would like to refer to the reply to first general comment.

**REFERENCES**

Crosbie, R. S., Binning, P., and Kalma, J. D.: A time series approach to inferring groundwater recharge using the water table fluctuation method, Water Resources Research, 41, 2005

Fahs, M., Younes, A., and Lehmann, F.: An easy and efficient combination of the Mixed Finite Element Method and the Method of Lines for the resolution of Richards' Equation. Environmental Modelling & Software 24.9: 1122-1126, 2009

Kavetski, D., Binning, P., and Sloan, S. W.: Adaptive time stepping and error control in a mass conservative numerical solution of the mixed form of Richards equation. Advances in Water Resources 24.6: 595-605, 2001

Kollet, S. J., and Maxwell, R.M.: Integrated surface–groundwater flow modeling: A free-surface overland flow boundary condition in a parallel groundwater flow model. Advances in Water Resources 29.7: 945-958, 2006

Pikul, M. F., Street, R. L., and Remson, I.: A numerical model based on coupled one-dimensional Richards and Boussinesq equations, Water Resources Research, 10, 295–302, 1974

**Response to review #3**

Referee comment on "Coupling saturated and unsaturated flow: comparing the iterative and the non-iterative approach" by Natascha Brandhorst et al., Hydrol. Earth Syst. Sci. Discuss., https://doi.org/10.5194/hess-2021-15-RC3, 2021

This article proposes two approaches for coupling saturated and unsaturated zones in a hydrological model. An explicit representation of the 3D flows is time consuming and a 1D approach to represent the unsaturated zone with a 3D approach to represent the saturated zone is an alternative which however raise the question of the representation the interface between these two environments which evolves over time. This lead, among other feature to variation in the specific yield. Two innovative approaches, one iterative and the other non-iterative, are presented in this article and compared to a 3D reference model.

We thank the reviewer for the effort and time to revise our manuscript and the constructive comments. We made clearer in the revised manuscript that the focus is on only one of the two approaches, while the other one is used as a fast, but more ad hoc reference approach (see also comments to the first reviewer). In the following, we will respond to all comments in detail.

General comments :

Even if the article is based solely on synthetic data, the presentation of the approach is clear and the results are convincing both in terms of the quality of the results and the efficiency of the calculation times. This makes it an interesting article that deserves to be published. My main concern is the conclusions on the lack of sensitivity of the unsaturated zone. To my point of view the constant monthly inflow may lead to a steady recharge of the saturated zone. It might be possible that with pronounced short term drying-wetting cycles, the role of surface parameter might be stronger, which could be amplified by the coupling between surface flux and soil water content due to transpiration regulation. In Figure 14 we can see that soil parameter have their strongest effect when changing the inflow regime (except the model warming period). I think the conclusion could be tempered on the prominent sensitivity of KGW

We thank the reviewer for this rating. Regarding the lack of sensitivity of the unsaturated zone, we understand the concern and agree that our conclusions were not general enough. In Fig. 14 (now 16) we see only a small influence of the unsaturated zone on the groundwater table position. This influence increases with larger depth to the groundwater table (not shown in the original version). We assume that this is because the unsaturated zone (especially the drier part close to the surface) has a stronger impact on the recharge under such conditions, which to us seems to be in line with the reviewer's argumentation. We now show this dependency in the revised manuscript (Fig.18) and made clear in the discussion that the dominance of KGW is specific for the flow conditions and there is a sensitivity of the unsaturated zone parameters under other conditions (drier soil, stronger impact on recharge), see l.489-500.

specif comments :

The shape of the equation is unusual. Considering that θ=S(hp)*Φ, I have difficulty to understand how d(θ)/dt lead to left member of equation 2 (θ being the volumetric soil water content). Can the author give reference. The specific storage Ss is not clear (dS/dhp ?)

The left member of Eq. 2 is derived from the time derivative in the volume balance (we assume incompressibility of water):

$$\frac{\partial V_w}{\partial t} = \frac{\partial (S(h_p)\phi V_t)}{\partial t} = \phi V_t \frac{\partial S(h_p)}{\partial t} + S(h_p)V_t \frac{\partial \phi}{\partial t} + S(h_p)\phi \frac{\partial V_t}{\partial t}$$

$$= V_t \cdot \left( \phi \frac{\partial S(h_p)}{\partial t} + S(h_p) \frac{\partial \phi}{\partial t} + \frac{1}{V_t} S(h_p) \phi \frac{\partial V_t}{\partial t} \right)$$

$$= V_t \cdot \left( \frac{\partial (S(h_p)\phi)}{\partial t} + S(h_p) \frac{\phi}{V_t} \frac{\partial V_t}{\partial h_p} \frac{\partial h_p}{\partial t} \right)$$

$$= V_t \cdot \left( \frac{\partial (S(h_p)\phi)}{\partial t} + S(h_p) S_s \frac{\partial h_p}{\partial t} \right)$$

with subscripts *w* and *t* denoting *water* and *total.*

The specific storage is in general defined as $S_s = \frac{1}{V_t} \frac{\partial V_p}{\partial h_p} = \frac{\partial \phi}{\partial h_p} + \frac{\phi}{V_t} \frac{\partial V_t}{\partial h_p}$ with $V_p = \phi V_t$ being the pore volume. This definition is used when the change of porosity is written in dependence of the change in water pressure head (so $\frac{\partial \phi}{\partial t} = \frac{\partial \phi}{\partial h_p} \frac{\partial h_p}{\partial t}$). One then also gets a slightly different formulation of Eq. 2: $\frac{\partial V_w}{\partial t} = V_t \cdot \left( \phi \frac{\partial S(h_p)}{\partial t} + S(h_p) S_s \frac{\partial h_p}{\partial t} \right)$.

In our case, we maintain the time derivative of porosity (or rather water content, as $S(h_p)\phi = \theta$) and therefore the specific storage formulation reduces to $S_s = \frac{\phi}{V_t} \frac{\partial V_t}{\partial h_p}$, which is equivalent to assuming that $\frac{\partial \phi}{\partial h_p} \cong 0$.

This formulation of Richards' equation can be found in e.g., Kavetski et al. (2001), Kollet and Maxwell (2006); Fahs et al. (2009). We understand the reviewer's comment that there are more common formulations of this equation and therefore, in the revised manuscript, clarified the definition of the specific storage, motivated our choice to use this formulation and gave a reference (now l.113-120).

L154 : How the saturation determined into the new cells (is water mass in the unsaturated layer preserved?

In the non-iterative approach, the saturation in the unsaturated cells is kept the same before and after resizing. In effect, this means that the amount of water in a cell is changing, as the saturation remains but the volume it relates to changes. We explain this in more detail in the revised manuscript (now l.163-166).

L162 1 ratio and three terms. Not clear

We agree that this is formally not correct. We give a ratio of two quantities here: the recharge or better the volume of water coming in from the unsaturated zone model ($R \cdot \Delta t_c$) and the volume of water added to (or subtracted from) the groundwater ($\phi \cdot \Delta H_{GW}$). Our formulation where we relate three terms was misleading here. We reformulated this in the revised manuscript (now l.175-178).

In Table 3 : Are KGW and KUZ conductivity at saturation? Here there is a decoupling of K Values. What happen in area that might belonging to the two domains.

Yes, they are the saturated hydraulic conductivity for the groundwater and the unsaturated zone model, respectively. In the revised manuscript we introduce them properly at their first appearance. In the area belonging to both domains, KGW is used as the overlapping part is always the saturated domain. We made this explicit as well (now l.367-369).

L393-396 Is the feature described here expected? May be can be addressed when discussing the yield values in part 4.5.

One cannot really say at this point what is expected because the heterogeneities have a strong impact which is hard to evaluate. Further analysis during the revision showed that there is a strong dependence on the soil properties at the water table position. We show

this in the revised manuscript (now l.452-458). A more detailed discussion on the Sy values is not possible without more tests.

Figure 12: errors are located in particular areas. Is there some explanation (boundary conditions, but not everywhere, heterogeneity patterns)?

Unfortunately, we made a mistake during the postprocessing of the data. The larger discrepancies occur close to the lower Dirichlet boundary (so between x=300m and x=400m). These differences are due to the zonation and the resulting simplified recharge pattern which hast a larger impact in this part of the domain. We corrected the figure and added another one (Fig.14) for explanation. We also adjusted the corresponding discussion part (now l.439-452 and 527-528).

L397 not clear what the ration is

We meant the ratio of the run times of the three compared models (two coupled and one fully integrated). This will be made clearer. We reformulated this part in the revised manuscript.

Section 4.4 Are the parameters leading to exfiltration (data removed ) cover particular domain, that might be meaningful

Yes, flooding occurs only for low values of the saturated hydraulic conductivity of the groundwater model KGW. Hence, the water table curvature needs to be higher to compensate for this, and therefore the groundwater table reaches the surface at the center of the domain. This is outlined in the revised paper (now l.466-467).

Fig 14-16 : is t[a] correspond to year unit (you may consider t[y]

We prefer to stick to the SI recommendation of using "a" for year.

**REFERENCES**

Fahs, M., Younes, A., and Lehmann, F.: An easy and efficient combination of the Mixed Finite Element Method and the Method of Lines for the resolution of Richards' Equation. Environmental Modelling & Software 24.9: 1122-1126, 2009

Kavetski, D., Binning, P., and Sloan, S. W.: Adaptive time stepping and error control in a mass conservative numerical solution of the mixed form of Richards equation. Advances in Water Resources 24.6: 595-605, 2001

Kollet, S. J., and Maxwell, R.M.: Integrated surface–groundwater flow modeling: A free-surface overland flow boundary condition in a parallel groundwater flow model. Advances in Water Resources 29.7: 945-958, 2006

---

## Author Response (AR2)

Dear authors,
Thanks for taking care of the reviewers comments and providing an improved manuscript.
After reading the revised version, I still have the following comments/questions which motivate minor revisions:

Dear editor,
We thank you for the effort and time to review our manuscript and our reply to the reviewers' comments. In the following we will answer to your comments in detail.

L145: The flux you apply is the real evapotranspiration. Should not be called potential evapotranspiration.

Yes, this is right. We removed the word "potential".

L221-223: Could you provide some guidelines to better qualify what is 'adequate' and 'sufficient' ?

We agree that this formulation is too imprecise. A convergence study on grid resolution and number of zones is needed to determine whether the grid is "adequate" and the number of unsaturated zone models "sufficient". We added "which can both be found by a convergence study" at the end of this sentence. With the convergence study one would refine the grid and the number of unsaturated zone models until results do not change more than a pre-defined criterion. We hope that the meaning of this is clear and therefore do not specify this further.

L268, typo 'Terefore'

Yes, right. We corrected that.

L366-369. Unclear to me. Does it mean that the heterogeneity of the unsaturated zone changes with time i.e. KUZ is replaced by KGW when the water level rises? Please clarify.

Yes, this is how it is done. The saturated cells of the unsaturated zone model are assigned the KGW value. This means that the conductivity changes when a cell becomes saturated or unsaturated. As we wanted to analyze both, the influence of the conductivity of the groundwater and that of the unsaturated zone, we had to choose: Either keep the K values in the cells fixed and get an inconsistency in the saturated (overlapping) part or change the K values in the unsaturated model according to the water table position. We decided to follow the second approach to avoid any possible numerical issues that could arise from large differences of K in the groundwater model and the saturated part of the unsaturated zone model. Besides, since we have (almost) hydrostatic equilibrium in the saturated part of the 1D models, the effect of changing K in those parts should be minor. We added the two sentences: "This means that the hydraulic conductivity of an unsaturated model cell is changed when the cell becomes saturated or unsaturated. Thus, inconsistencies between the two model compartments that may lead to numerical problems are avoided." for clarification.

Sensitivity analyses
You have to convince me that 316 simulations and a second order polynomial are enough to described f(x) (equation 16) for your highly coupled non-linear models... The simplest way to do this is to perform the polynomial fitting without taking care of some model outputs (say about 20 outputs) and then compare the predicted values with the model outputs for these 20 values (cross validation). A nice plot with the differences between both will show the quality of f(x).

We did such a plot for all three observation types (water table position, water table fluctuation and specific yield). We chose the last time step to be sure that no spin-up

effects tamper with the result and used 26 samples for validation. The result is shown in Figure 1. For the water table fluctuations, the fitting is less accurate with $R^2$ at 0.81, but for the other two observation types, the model output and the approximated output fit very well. We added the sentence "The polynomial representation (Eq. 16) was cross validated with 26 randomly chosen parameters and found to be appropriate (result not shown)." at the end of the first paragraph in Section 4.4. We can also add the plot to the manuscript but prefer to keep it out as it is no central point and would make the manuscript even longer.

[Figure]

Figure 1: Observations and observations approximated by f(x).

You nicely showed that the iterative algorithm is more accurate. What is the interest of performing the sensitivity analyzes with the non-iterative scheme? The physics embed in the model is the same and the model outputs for both scheme very close. Therefore, L529 is obvious. What about removing the non-iterative scheme in the GSA?

This is a good suggestion. We removed the non-iterative coupling scheme from the GSA.